# The Therapeutic Value of Solanum Steroidal (Glyco)Alkaloids: A 10-Year Comprehensive Review

**DOI:** 10.3390/molecules28134957

**Published:** 2023-06-23

**Authors:** Julien A. Delbrouck, Michael Desgagné, Christian Comeau, Kamal Bouarab, François Malouin, Pierre-Luc Boudreault

**Affiliations:** 1Institut de Pharmacologie, Faculté de Médecine et des Sciences de la Santé, Université de Sherbrooke, 3001 12e Avenue Nord, Sherbrooke, QC J1H 5N4, Canada; julien.delbrouck@usherbrooke.ca (J.A.D.); michael.desgagne@usherbrooke.ca (M.D.); christian.comeau@usherbrooke.ca (C.C.); 2Centre SEVE, Département de Biologie, Faculté des Sciences, Université de Sherbrooke, 2500 Boul de l’Université, Sherbrooke, QC J1K 2R1, Canada; kamal.bouarab@usherbrooke.ca; 3Département de Biologie, Faculté des Sciences, Université de Sherbrooke, 2500 Boul de l’Université, Sherbrooke, QC J1K 2R1, Canada; francois.malouin@usherbrooke.ca

**Keywords:** *Solanum*, steroid, glycoalkaloid, alkaloid, secondary metabolites, bioactivity

## Abstract

Steroidal (glycol)alkaloids S(G)As are secondary metabolites made of a nitrogen-containing steroidal skeleton linked to a (poly)saccharide, naturally occurring in the members of the Solanaceae and Liliaceae plant families. The genus *Solanum* is familiar to all of us as a food source (tomato, potato, eggplant), but a few populations have also made it part of their ethnobotany for their medicinal properties. The recent development of the isolation, purification and analysis techniques have shed light on the structural diversity among the SGAs family, thus attracting scientists to investigate their various pharmacological properties. This review aims to overview the recent literature (2012–2022) on the pharmacological benefits displayed by the SGAs family. Over 17 different potential therapeutic applications (antibiotic, antiviral, anti-inflammatory, etc.) were reported over the past ten years, and this unique review analyzes each pharmacological effect independently without discrimination of either the SGA’s chemical identity or their sources. A strong emphasis is placed on the discovery of their biological targets and the subsequent cellular mechanisms, discussing in vitro to in vivo biological data. The therapeutic value and the challenges of the *solanum* steroidal glycoalkaloid family is debated to provide new insights for future research towards clinical development.

## 1. Introduction

Nutrition is a vital act of communion with nature. Plants use the sun’s energy to transform minerals and gases into a cornucopia of organic molecules such as carbohydrates, lipids, terpenes, steroids, etc. Directly or indirectly, they supply all the energy and building blocks needed for the growth and maintenance of the human body. History shows that plants have been used for their medicinal properties in all civilizations and they continue today to be a source of bioactive molecules. The *Solanum* genus is comparatively a large one, encompassing over 1500 species and it is of particular importance in most of the world, as it includes popular food crops such as tomato (*Solanum lycopersicum*), potato (*Solanum tuberosum*) and eggplant (*Solanum melongena*). Two etymologies have been proposed: *sol* + *anum* (“of the Sun”) or *solor* + *nus* (“soothing”). Fittingly, it also contains various nightshade species used as folk medicine [1], such as the yellow-fruit nightshade (*Solanum virginianum*) used in Ayurveda medicine and the black nightshade (*Solanum nigrum*) used in both Western and Eastern traditional medicine. The genus *Solanum* is part of the Solanaceae family, which also includes the genera *Capsicum* (bell and chili peppers), *Nicotiana* (tobacco) as well as *Datura*, *Mandragora* and *Atropa* among others. It can thus be seen that many plants of this family are cultivated for their secondary metabolites like capsaicin and nicotine or are poisonous such as the deadly nightshade (*Atropa belladonna*).

At the interface between food, medicine, and poison, Steroidal Alkaloids (SAs) and their glycosidic versions are secondary metabolites that are especially prominent in the *Solanum* genus. Steroidal GlycoAlkaloids (SGAs) consist of a (poly)saccharide linked to a hydrophobic steroidal skeleton containing a nitrogen atom. They accumulate in different organs of the plants (leaves, roots, fruits, tubers, etc.) [2,3,4] and play a key role in the plant-environment interactions, in particular against bacterial and fungal attacks [5,6]. Their anti-nutrient properties and human toxicity are well-documented [3,7,8,9,10,11], while they more recently have attracted the attention of researchers for their wide range of pharmacological properties (anticancer, antibiotic, anti-inflammatory, etc.) [12,13,14,15]. For instance, Coramsine (SBP002) was an experimental chemotherapeutic drug composed of two steroidal glycoalkaloids (solamargine and solasonine) isolated from the species *Solanum linneanum* (devil’s apple) [16].

Several reviews are focused on either the discovery, characterisation, natural occurrence, nutritional properties, toxicities or bioactivities of steroidal glycoalkaloids [12,17,18,19,20,21,22,23]. Instead, we provide herein a unique comprehensive review of the pharmacological targets and activities of *Solanum* S(G)As to accentuate their interest as health promoters. 

## 2. Methodology

The methodology used to analyze scientific reports dealing with *Solanum* SGAs and their bioactivities is summarized in Figure 1. Nine separate Pubmed queries were made on 14 October 2022, using the following keywords: “*Solanum* Steroid Alkaloid”, “Solan* Steroid Alkaloid”, “Tomato Steroid Alkaloid”, “Potato Steroid Alkaloid”, “Eggplant Steroid Alkaloid”, “Aubergine Steroid Alkaloid”, “Tomatidine”, “Solamargine” and “Solanine”. The literature research was limited to the last 10 years (2012–2022; *n* = 914). Upon duplicate removal, titles, and abstracts *(n* = 460) were closely read to determine their relevance towards human health benefits (*n* = 209). Among the remaining pertinent articles (*n* = 130), reports dealing with plant extract or an unidentified mixture of bioactive compounds were excluded (*n* = 41). A deep analysis of the relevant articles (*n* = 89) was carried out to focus on the bioactivities of *Solanum* S(G)As and their mechanisms of action, with the goal to unveil their values and challenges towards the development of new therapeutic treatments (Table 1). In addition, the applicable literature debating the classification, occurrence and toxicology of *Solanum* SGAs has been used to complement the writing of this review article without time limitation. The anti-cancer properties of *Solanum* SGAs have been thoroughly reported over the last few decades in exhaustive review articles. Therefore, scientific articles reporting the cytotoxic properties of *Solanum* SGAs have been ruled out of discussion but nonetheless summarized in Table 2 (*n* = 79).

## 3. Steroidal (Glyco)Alkaloids: Classification

Steroidal alkaloids are nitrogenous derivatives of steroids and have been isolated and characterized from many organisms including animals (amphibians, sea sponges, etc.) and plants (Solanaceae, Liliaceae, etc.). They are important secondary metabolites with a wide range of potential therapeutic applications [145]. For instance, the steroidal alkaloid drug marketed Zytiga^®^, was approved in 2011 by the FDA for the treatment of metastatic castration-resistant prostate cancer [20]. The nitrogen atom is incorporated to the steroidal backbone either as heterocyclic ring, basic side chain or as a substituent at the C-3 position. Monomeric steroidal alkaloids are commonly classified into four groups based on their carbon heterocyclic skeleton: (1) Pregnane (C_21_); (2) Cyclopregnane (C_24_); (3) Cholestane (C_27_); (4) Others (Figure 2). Comprehensive reviews on the classification of steroidal alkaloids have been published and do not enter in the scope of this review article that focuses on *Solanum* S(G)As [20,146,147].

The occurrence of the *Solanum* steroidal alkaloid is well-documented and they mainly belong to C_27_-cholestane family [20,21]. These metabolites are characterized by the common ABCD steroid skeleton, and they are conventionally subdivided into three main types: **spirosolane**; **solanidane** (aka. Solanidine); and **verazine** (aka. 22/23,26-epiminocholestane) types (Figure 3). 

The **Spirosolane** type is characterized by a 1-oxa-6-azapiro [4,5] decane ring system (E and F rings) and can be divided into two stereoisomeric subgroups: 22αN- and 22βN-spirosolanes (Figure 3). The C_27_ methyl group always take place in equatorial position within the Spirosolane family. The **Solanidane** family possess a fused indolizidine pattern with the 22R,25S configuration (Figure 3). Solanidine and its 5a,6-dihydro analog, demissidine, are the most common SAs of this structural entity. The **Verazine** group has a 22/23,26-epiminocholestane skeleton and differs by the non-conjugated N-containing F-ring and the absence of ring E (Figure 3). For analogs with a saturated piperidine F-ring, 22αN and 22βN-diastereoisomers also exist. Noteworthy, the C_16_ position can be saturated or substituted (e.g., -OH).

Over the last few decades, purification and extraction techniques of natural compounds have considerably progressed along with the abilities of researchers to elucidate complex chemical structures. Therefore, further phytochemical investigations resulted in the discovery of new metabolites with unique structural features thus leading to the creation of six additional (sub)types: **α-Epiminocyclohemiketal**; **Leptine**; **23,26-Epiminocholest-23(N)-en-22one**; **3-Aminospirostane**; **26-(N)-Acylaminocholestan-22-one**; and **16-Aminocycloxy-pregnenolone** (Figure 3) [20,21].

The most prevalent metabolite of the **α-Epiminocyclohemiketal** family is solanocapsine, which contains two nitrogen atoms and is characterized by its α-epiminocyclohemiketal functionality. Its worth mentioning that 3-hydroxy analogues were discovered later. **Leptines** are solanidane equivalents with a 23-hydroxy/acetyl group. For instance, leptinidine is the 23β-hydroxy analog of solanidine (Figure 3). The **23,26-Epiminocholest-23(N)-en-22-one** type is defined by the C_22_ exocyclic carbonyl function and the unsaturated bond between the C_23_ and the nitrogen of the five-membered ring. The **3-Aminospirostane** group (aka jurubidine type) is constituted of 3-amino analogues of spirostane steroids that contrast from previous metabolites by the oxygen replacing the basic nitrogen N_22_. Another family characterized by the lack of N_22_ basic nitrogen is the **26-(N)-Acylaminocholestan-22-one**. These metabolites exhibit an almost identical side chain to the cholestane framework, likely because the N-acetyl group prevents the cyclization. The **16-Aminoacyloxy-pregnenolone** type is unique with its C_21_ pregnane skeleton instead of the usual C_27_ cholestane framework. It is a recent category with only two SGAs isolated to date [148,149].

It is to be noted that steroidal alkaloids often exist in nature as glycosides named Steroidal GlycoAlkaloids (SGAs) [150]. The biosynthesis of *Solanum* SGAs has been partially elucidated and it relies on the cytosolic mevalonate pathway with cholesterol as a key precursor [151,152]. It can be separate into two main operations: the aglycone construction and glycosylation. A significant number of glycoalkaloid metabolism (GAME) genes are involved in the initial regulation of the structural modification of the cholesterol backbone by hydroxylation, transamination, oxidation, etc., followed by glycosylation reactions [153].

The polar entity in SGAs includes one to several monosaccharides in various combinations (branched, linked or linear). The most common units are D-galactose, D-glucose, D-xylose, L-arabinose and L-rhamnose which interconnected offer a variety of disaccharides, trisaccharides and tetrasaccharides [21]. The predominant glycosidic chains of the solanum SGAs are solatriose, chacotriose, lycotetraose and commertetraose (Figure 4). The glycosidic linkage mostly occurs at the C_3_ position of the steroidal backbone (monodesmosidic) by contrast to bisdesmodic metabolites that own an additional sugar at another location of the steroidal skeleton (e.g., Esculeoside A) [154]. The α/β anomeric configuration of the glycosidic bond at the C-3 hydroxy position of the aglycone is driven by the D/L-configuration of the monosaccharide. The oligosaccharide chain plays a pivotal role in directing biological applications as observed for the SGA Tomatine and its non-glycosidic analog Tomatidine that display different therapeutical properties (see Section 5).

The popularity of SGAs arises from the structural diversity offered by the aglycone structure, the nature of the side chain and the combination of carbohydrates. The aglycone unit often shares the tetracyclic 5α-androstane of the C_27_ cholestane skeleton while divergence mostly occurs by substitutions; rearrangement of the side chain (C_20_–C_27_); and the position of substituents. Interestingly, 5α-saturated and 5-unsaturaded analogs have been detected for almost all categories of *Solanum* SGAs.

## 4. Occurrence of *Solanum* SGAs

**Spirosolane** and **Solanidane** SGAs display the highest structural diversity and they are the major representative families of SGAs within the *Solanum* genus [17,155]. They are characterized by a five-membered E-ring and a six-membered F-ring. The chemical structures of the most popular *Solanum* SGAs are depicted in Figure 5. α-solanine and α-chaconine are potato metabolites known for their well-established toxicity. Their productions are affected by genetics and by numerous factors such as temperature, light, storage conditions, etc. For instance, potatoes left exposed to the sun turn green and are inedible because of their increased amounts of SGAs [156,157,158]. α-tomatine and its C_5_/C_6_-unsaturated analog (dehydrotomatine) are the main SGAs responsible for the bitter taste of immature green tomatoes. α-tomatine accumulates in all parts of the plant with a maximal concentration up to 500 mg/kg in fresh green tomatoes [7,159]. This metabolite is converted into esculeoside and lycoperoside upon maturation of the fruit while it can also be found in several other *Solanum* species [160,161,162]. α-solasonine and α-solamargine are characteristic SGAs of aubergine species and mostly accumulate in the flesh [163]. To date, over 100 structurally unique SGAs have been isolated from *Solanum* species and attract the focus of scientists because of their wide range of pharmacological applications [17,18,164].

## 5. Therapeutic Context—*Solanum* S(G)As

*Solanum* SGAs and their aglycones have been extensively researched for their therapeutic applications in the last ten years. Several reviews have regrouped their biological activities [17,18,19,20] but mainly on the side of cytotoxic and/or cancer-related applications [13,165,166,167,168]. As such, this review will not target the cytotoxic activities of SGAs unless the data is included in a research article specifically researching other biological effects (i.e., LD_50′_s on specific cell lines relevant to the studied model). However, a table for cancer applications and references will still be included later in this review as reference (Table 2). The prior art on *Solanum* SGAs is wide and well detailed but lacks in consistency on the therapeutic applications, whereas most reviews target specific secondary metabolites and only scrape the surface of the well documented pharmacological effects. This review therefore integrates the various biological effects of *Solanum* SGAs without discrimination as all molecules were included, without fail, over a 10-year period. Moreover, to our knowledge, it is the first review which tackles precise mechanisms of action of all SGAs in a single manuscript. Accordingly, we herein provide a unique comprehensive overview of the therapeutic value of *Solanum* SGAs over the last ten year. A strong emphasis is put on the identification of SGA’s biological targets, the discovery of the cellular mechanisms and the discussion of pharmacological data from in vitro to in vivo results.

### 5.1. Antibiotic

In studies on the antibiotic properties of α-tomatine (α-TN) and its aglycone derivative tomatidine (TO), it was found that only TO could show specific growth inhibition of small-colony variants (SCVs) of *Staphylococcus aureus* [24,25]. SCVs are clinical strains associated with persistent and chronic infections and their slow-growth phenotype is often associated with mutations in components of the respiratory chain (e.g., mutations in genes responsible for the biosynthesis of heme or menadione) leading to a reduction in ATP production. Normal phenotypes of *S. aureus* were not inhibited by the presence of TO, whereas SCVs were inhibited at extremely low concentrations of TO (minimal inhibitory concentration [MIC] of 0.06–12 µg/mL). TO action was linked to the electron transport system as 4-hydroxy-2-heptylquinoline-N-oxide (HQNO)—an inhibitor of the electron transport system—sensitized the non-SCV strains of *S. aureus* to TO. TO was later defined as a bacteriostatic molecule against SCVs. Interestingly, although TO had no inhibitory effect on the normal-growth strains of *S. aureus*, it was shown to improve up to 16-fold the antibiotic efficacy of several aminoglycosides or cyclic peptides in MIC assays on several strains from diverse origins [25,26].

The synergistic effect of TO with several antibiotics against various bacterial species has been further studied by Soltani [27]. As previously demonstrated, TO showed a significant potentiating effect for gentamicin against *S. aureus* but these authors also suggested that some effect of TO could be observed for ampicillin against *E. faecalis* and for cefepime and ciprofloxacin against *P. aeruginosa*. However, in such cases, the calculated Fractional Inhibitory Concentration index (FIC) for the detection of synergy was near or at 0.5, indicating a phenomenon of additivity rather than a true synergy. None of the antibiotics used in combination with TO showed a synergistic effect against *Escherichia coli*.

Medicinal chemistry on TO was performed later to find a more potent compound. However, on the 55 compounds reported by Chagnon et al. [28], no compound managed to inhibit growth of *S. aureus* SCV variants as potently as TO. Some of the modifications which were tested are: the formylation of the amine, inversions of the stereogenic center of the alcohol, alkylations, oxidations of the alcohol moiety, replacement of the alcohol by alkyl diamines, F-ring opening and several replacements of the amino ketal moiety. Although TO-related compounds alone did not manage to inhibit growth as potently, some did have the same inhibitory effects as TO when used in conjunction with gentamicin.

In addition to the bacteriostatic potential of TO toward *S. aureus*, it also exhibits bactericidal effects when *S. aureus* is in co-culture with *P. aeruginosa* [29]. The authors describe the effect by showing that a TO/tobramycin combination was able to kill *S. aureus* in the presence of *Pseudomonas aeruginosa* by interspecific small-molecule interactions. Firstly, the authors showed that *S. aureus* was killed in a co-culture using TO at the dose of 8 µg/mL, whilst *P. aeruginosa* was not. Interestingly, *S. aureus* was not killed in a monoculture by the same concentration of TO. Boulanger and associates showed, using HQNO-deficient mutants of *P. aeruginosa*, that HQNO was one of the factors potentiating the anti-staphylococcal effect of TO. Addition of pure HQNO in *S. aureus* monocultures also potentiated the effects of TO, creating a bacteriostatic medium. Furthermore, combinations of tobramycin and TO were able to kill methicillin-resistant *S. aureus* and *P. aeruginosa* in co-culture, whilst neither antibiotic was able to kill any bacterium by itself. 

From the previously published structure-activity relationship study (SAR) of TO, one compound was investigated further as it did exhibit bactericidal activity against prototypic strains of *S. aureus* by itself, in contrast to TO [30]. The ethylene diamine moiety of FC04-100 provided activity against both prototypic *S. aureus* and SCVs, and preserved its synergic activity when used in combination with aminoglycosides. TO (8 µg/mL) was bactericidal against *S. aureus, Bacillus subtilis* and *B. cereus* when used in conjunction with gentamicin at doses ranging from 0.06–2 µg/mL, whereas only the combination with FC04-100 was able to kill *Listeria monocytogenes* significantly after 24 h. FC04-100 also managed to kill significantly more prototypical *S. aureus* embedded in biofilm than gentamicin alone, with 3.2 to 3.8-log differences depending on the dose of gentamicin (0.12 to 0.5 µg/mL).

TO’s molecular target was only later identified by Lamontagne Boulet et al. [31] The authors used TO as a selective pressure on *S. aureus* SCVs in order to create resistant strains and harvested mutants with more than 3-log improved resistance to TO. The mutations were identified in the gene ccpA (key metabolic regulator in low-GC Gram-positive bacteria) for a first level of resistance and in atpE (ATP synthase subunit C) for full resistance. The research group later confirmed that TO and FC04-100 targeted the ATP synthase of *S. aureus* via genetic manipulations of the bacterium, and they created a homology model of the ATP synthase and more precisely of the subunit C. The similarity of the ATP synthase subunit C among Bacillales (Staphylococcus, Bacillus, Listeria) explained the selectivity of TO action toward these bacterial species. An ATP synthase inhibition assay was put in place using inverted bacterial membrane vesicles and was used to correlate the binding affinity of several compounds with their antibiotic potency. The hypersusceptibility of SCVs to TO (as opposed to that seen with prototypical *S. aureus* strains) was thus explained by the already impaired respiratory chain of SCVs, resulting in a critically lower ATP production in the presence of TO or FC04-100.

After the identification of the target, Langlois and associates attempted to further explain the adjuvant effect of TO on aminoglycoside activity against prototypical *S. aureus* strains [32]. In this paper, the authors first demonstrated a link between the level of susceptibility to TO and the relative hydrophobicity of the bacterial cell. The authors explain that the highly lipophilic TO would interact in a lesser way with less hydrophobic surfaces. As for studies on the bactericidal action, the authors showed that the TO-gentamicin combination impacted the membrane potential and generated reactive oxygen species. TO was also responsible for the sensitization to gentamicin in prototypical strains as gentamicin uptake was increased by more than 60% in the presence of TO over a 2 h time course. Finally, even when used alone, TO was shown to have a great impact on prototypical *S. aureus* when grown in anaerobic conditions, reminiscent of its action on respiratory-deficient SCVs. 

In another study [33], the anti-biofilm properties of TO were investigated against *L. monocytogenes.* TO was able to reduce formation, volume and thickness of the biofilm at a dose of 144 µM against some strains. However, the reduction in the biofilm formation was not attributed to the growth inhibition. The reduction in the biofilm formation was rather explained by the reduction of cell adhesion. Interestingly, TO was also shown to reduce *L. monocytogenes* cell motility. These non-growth inhibitory actions of TO on biofilm formation and cell motility in *Listeria* are similar to the action of TO on the virulence attributes of prototypical *S. aureus* (e.g., the reduction of alpha-hemolysin production), most likely due to the reduction in ATP production in the presence of TO, although a direct effect of TO on bacterial membranes and membrane-associated cell signaling (e.g., quorum sensing) cannot be ruled out.

Another steroidal alkaloid, Solanindin B, obtained by acidic hydrolysis of SGAs Solanindioside A-C, isolated from the fruit *Solanum violaceum*, was able to suppress *S. aureus* growth by 50% and >99% at 37 and 200 µM, respectively [91]. Although this activity against prototypical *S. aureus* was weak, it would be interesting to see if such compound could be as efficient as TO against SCVs. Noteworthy, the double bond at C_3_, C_4_ of the steroidal skeleton proved pivotal for the inhibitory activity, thus demonstrating the interest of additional structural-activity relationship studies to enhance biological potency. 

The impact of the steroidal glycoalkaloid Solamargine (SM) on *Pseudomonas aeruginosa* biofilm formation and pyocyanin production has been studied by Cabanillas [81]. Pyocyanin is a virulence factor that can cause oxidative stress, damage host cells and kill bacterial competitors of *P. aeruginosa*. It can be said that although growth inhibition of *P. aeruginosa* H103 was not directly measured, a reduction of about 20% of the biofilm formation and pyocyanin production in the presence of 50 µg/mL of Solamargine in vitro may imply that such a compound could moderately affect virulence during host-pathogen interactions.

### 5.2. Antifungal

In addition to the antibacterial effects described above, Solamargine was also reported to have a moderate antifungal activity against *Trichophyton mentagrophytes* and *C. albicans* (MIC 64 µg/mL) [81]. In fact, as described below, a few other SAs and SGAs were shown to have antifungal activities.

Solanopubamine (SPB) isolated from *Solanum schimperianum* has been shown to inhibit growth of pathogenic yeast species [68]. In MIC assays against several species of fungi and bacteria, SPB showed an MIC of 12.5 µg/mL for *Candida albicans* and *C. tenuis*, while showing minimal activity against *C. krusei*, *C. neoformans*, *C. glabrata*, *Aspergillus fumigatus*, *S. aureus*, *E. coli*, *P. aeruginosa* and *Mycobacterium* intracellulare. The authors investigated methylations, acetylations and couplings with either octadecanoic or undec-11-enoic acids on the 3-amino group or on the C-23 hydroxyl function of SPB. All modifications had negative impact on MIC values. Moreover, substitution of the alcohol moiety by an acetyl group completely negated the antifungal activity of SPB, which supposes that either the H-bond donor of the alcohol moiety is important, or the steric hindrance around the oxygen might hinder activity. Any modification of the amine on position three yielded poor results in MIC assays.

Besides the antibacterial activity reported earlier, TO has also been shown to have fungistatic activity [34]. The author from this article screened several natural products (NPs) in MIC assays on *C. albicans* at either pH 7 or 4.6, simulating the different niches of this pathological fungi. Several NPs have shown moderate activity with the most potent being TO. TO was subsequently tested in MIC against *C. krusei*, *C. tropicalis*, *C. parapsilosis* and *C. glabrata*, with all strains except *C. glabrata* being highly susceptible to TO, even showing better MIC’s than Fluconazole especially against *C. krusei*. This activity of TO was unexpected, as previous studies generally recognized that α-Tomatine (α-TN) and not TO, its aglycon version, had an inhibitory activity against phytopathogens and could disrupt eukaryote cell membranes by interacting with ergosterol and cholesterol [1,2]. Indeed, to counter the antifungal activity of TN, phytopathogen fungi, most of them being solanaceous pathogens, produce TN-modifying enzymes to compromise the antimicrobial activity of TN by acting on the lycotetraose part of the molecule leading to the release of less antifungal intermediate compounds, including TO [169,170,171,172]. The tomato leaf spot fungus *Septoria lycopersici* produces a TN-modifying enzyme that hydrolyzes the terminal β-1,2-linked D-glucose molecule from α-TN to give β2-TN (Osbourn et al., 1995). The grey mold disease agent *Botrytis cinerea* releases a TN-modifying enzyme which acts on the the terminal β-1,3-linked D-xylose molecule from α-TN to release β1-TN [169]. *Fusarium oxysporum* produces a tomatinase which releases the lycotetraose from TN to give TO [173]. The compromised inhibitory effect of these intermediate degradation products in comparison to TN is not general on phytopathogen fungi, including the ones that produce TN-modifying enzymes [169].

The precise mode of action of TO was, however, explored by several groups [43,50] and identified TO as an ergosterol biosynthesis perturbator, more precisely targeting the C-24 sterol methyltransferase Erg6, which would explain the susceptibility of fungal strains to TO and the low cytotoxicity of TO toward mammalian cells (absence of Erg 6). Dorsaz et al. [34] also suggested an inhibition of Erg4 (C-24 sterol reductase) by studying the transcriptional response and sterol content of *C. albicans* after incubation with TO. Finally, the in vivo efficacy of TO in a murine *C. albicans* systemic infection model was also investigated, and intraperitoneal injections of TO showed significant reduction of *C. albicans* in kidneys 48 h post-infection [34]. 

Solasodine-3-*O*-β-D-glucopyranoside (SG), isolated from the plant *Solanum nigrum L.*, was also studied for its antifungal properties [79]. SG exhibited anti-adhesive properties in adhesion models of *C. albicans* at doses as low as 8 µg/mL. Moreover, SG shows inhibition of yeast-to-hyphea transition, biofilm formation and even improves nematodes (*C. elegans*) survival after *C. albicans* infection at the dose of 8 µg/mL, while exhibiting minimal toxicity on the nematodes at 64 µg/mL. However, SG was slightly toxic on human cell lines in the MTT assay, with IC_50′_s on PC3 and HBE cells of 33.59 and 41.4 µg/mL, respectively. SG has also been described as a prodrug by Chang et al. in a recent paper [80]. The authors show that solasodine is highly effluxed by comparing accumulation of solasodine in wild type and efflux-deficient strains of *C. albicans*. The authors also showed that SG was hydrolysed in solasodine, and that this hydrolysis is suggested to appear at the cytoplasmic membrane of *C. albicans*. 

The moderate antifungal activity of Solamargine against *T. lentagrophytes* and two strains of *C. albicans* (MIC 64 µg.mL^−1^ towards all strains) has been reported by Cabanillas in 2021 [81]. 

### 5.3. Anti-Protozoal/Anti-Parasitic

Leishmaniosis is a disease induced by kinetoplastid protozoan parasites. Solamargine and solasonine (*Solanum lycocarpum*) have both shown interesting antiparasitic activities. In 2013, Miranda et al. [83] reported in vitro leishmanicidal activity for solamargine and solasonine on promastigote forms of *Leishmania amazonensis*. The equimolar mixture of both SGAs interestingly displayed a synergistic potency after 72 h (IC_50_ = 1.1 µM) compared to individual solamargine (IC_50_ = 6.2 µM) and solasonine (IC_50_ = 7.8 µM). Solasodine, the aglycone unit of both SGAs, displayed significantly lower activity (IC_50_ = 219 µM) indicating the importance of the sugar moieties. This result highlighted the pivotal role of the carbohydrate chain in the inhibition mechanism. Besides, cytotoxicity assays were performed on LLCMK_2_ cells in order to determine a selectivity index (SI = IC_50_(LLCMK_2_ cells)/IC_50_(protozoa)). The equimolar mixture of both SGAs displayed mild cytotoxicity after 72 h (IC_50_ = 10.1 µM) similar to solamargine alone (IC_50_ = 11.2 µM) while solasonine was slightly less toxic (IC_50_ = 20.6 µM). The highest selectivity index (SI = 9.1) was therefore obtained for the equimolar mixture of both steroidal glycoalkaloids in this antiparasitic model. 

In 2016, Lezama-Dàvila and co-workers further investigated the leishmanicidal activity of combined solamargine and solasonine against *L. mexicana* promastigotes in cellulo [84]. Solamargine and Solasonine had an IC_50_ of 35 and 36 µM after 48 h and a therapeutic window of 9 and 20 respectively, showing little toxicity on dendritic cells. In vivo, ear lesions infected with amastigotes of *L. mexicana* treated with 10 µM of solamargine and solasonine per mouse per day showed a reduced progression compared to the negative control. Similarly, the number of parasites was also reduced. Interestingly, this quantity of steroid was below the reported in vitro IC_50′_s but still managed to create the desired effect. Both metabolites also showed significant induction in the expression of NF-κB/AP-1 in RAW 264.7 macrophages. Solamargine showed a significant decrease in nitric oxide (NO) production in these cells. Cytokine quantities were also evaluated in LPS-induced macrophages and dendritic cells, where solamargine showed significant reduction in TNF-α and significant increases in IL-6. These results demonstrated the efficacy of topical preparation of solamargine and solasonine to suppress cutaneous leishmaniasis in C57BL/6 mice. 

The anti-parasitic activities of solamargine and solasonine against various flagellated protozoa has been reported. Previous reports suggest that both SGAs may display a synergistic effect when used in conjunction against parasites. This ability was confirmed by Moreira et al. [85] when they investigated the effect of solamargine and solasonine on *Giardia lamblia*, a protozoon responsible for a type of intestinal infection (giardiasis). As previously observed, the mixture of both SGAs displayed higher potency (IC_50_ = 13.2 µg/mL) than individual solamargine (IC_50_ = 120.3 µg/mL) and solasonine (IC_50_ = 103.7 µg/mL), thus demonstrating a synergistic effect. Surprisingly, the cytotoxicity assay on macrophage cells showed significantly higher toxicity for solasonine (IC_50_ = 62.5 µg/mL) versus solamargine (IC_50_ > 2000.0 µg/mL). The selectivity index (SI; IC_50_ (macrophage cells)/IC_50_ (protozoa)) of the equimolar mixture was about 19.

The anti-protozoal effect of various S(G)As on three strains of mucosal Trichomonads, protozoan parasites known to cause diseases in humans, cattle and cats have been studied by Liu and Friedman [66]. A general screening of the inhibitory activity at 100 µM for several natural bioactive compounds has been conducted against three strains: *Trichonomonas fetus feline C1*, *T. fetus bovine D1* and *T. vaginalis human G3*. Noteworthy, tomatine, α-solanine and α-chaconine completely inhibited all the strains’ growth at such a concentration, while their respective aglycone tomatidine and solanidine showed low to moderate inhibitory activities (<50%). These results highlighted the importance of the glycosidic chain towards the antiprotozoal activities. The potency of tomatine, α-solanine and α-chaconine was further evaluated at various concentrations. Tomatine showed the most active metabolite with IC_50_ values of 2, 3 and 8 µM against *C1*, *D1* and *G3,* respectively. Although the potency of tomatine was lower than the current drug treatment metronidazole (IC_50_ < 0.75 µM for all strains), it did not affect bacteria of the normal flora (*Lactobacillus* and *E. coli*) unlike most antibiotics. Again, this showed that the glycosylated and aglycon version of SAs result in different antimicrobial activities and selectivity spectra.

The trypanocidal activity of solamargine has been reported by Moreira and co-workers using a MTT colorimetric assay to measure the viability of *Trypanosoma cruzi* (Y strain) [86]. This protozoan is responsible of the Chagas’ disease, a major public health problem especially in Latin America. Solamargine displays an IC_50_ value of 15.3 µg/mL on *T. cruzi*, very close to the first line treatment benznidazole (Bz, IC_50_ 9.0 µg/mL). In 2021, the synergistic effect of the steroidal alkaloid Tomatidine (TO) with Posaconazole against the protozoan *Trypanosoma cruzi* was demonstrated in vitro and in vivo [49]. Tomatidine displayed a lower-to-similar EC_50_ value (0.78 µM) than benznidazole (EC_50_ = 2.22 µM) against *T. cruzi* intracellular amastigotes (Tulahuen C2C4 strain). However, TO did not exhibit antiparasitic activity in vivo, likely because of its moderate solubility in aqueous media. Accordingly, TO was evaluated in vitro for drug combination with the reference drug (Bz), the azole-type CYP51 inhibitor Posa and Fexinidazole. The synergistic effect was only observed for TO and Posa association with the strongest effect for a 3:1 ratio, respectively (FIC index 0.1). The reduction of parasitemia and mortality was subsequently assayed in vivo in a mouse model of acute *T. cruzi* infection. The co-administration of TO and Posa (ratio 3:1, 5 mg/kg) induced a significant reduction of parasitemia (80%) and increased the animal survival by 60%. It strongly supports the synergistic effect between the two molecules, knowing that Posa (10 mg/kg) and TO (5 mg/kg) alone displayed only 40% and 0% of survival rate compared to the control. The synergy may be explained by both compounds inhibiting different enzymes of the sterol biosynthetic pathway (lanosterol 14α demethylase and sterol 24-C-methyltransferase), thereby impacting metabolism of *T. cruzi* at two different checkpoints.

Over the last 10 years, three SGAs: β-solamarine; solamargine; and solasonine have been reported to display molluscicidal and/or schistosomicidal activities. In 2015, Njeh et al. highlighted the in vitro molluscicidal activity of β-solamarine against *Galba truncatula Müll*., the snail intermediate host of the parasite *Fasciola hepatica L.*, causing fascioliasis infection [96]. β-solamarine can be found in various *Solanum* species and showed a median lethal concentration (LC_50_) value of 0.49 mg/mL against *Galba truncatula* after 48 h of treatment. Later, Alsherbiny and co-workers reported the activity of solasonine and solamargine against *Schistosoma* worms and their intermediate, host snails [87]. Schitosomiasis is currently the second most prevalent parasitic infection after malaria and molluscicides are the usual method of choice to eradicate the disease. Solamargine and solasonine a showed similar efficacy (LC_50_ ~10 ppm) against *Biomphalaria alexandrina* snails whereas the aglycone skeleton solasodine was significantly less active (LC_50_ = 45.5 ppm). Likewise, the schistosomicidal potency against *Schistosoma mansoni* is almost identical for both SGAs (LC_50_ ~7 ppm) and lower for the aglycone solasodine (LC_50_ >50 ppm). 

### 5.4. Antiviral

Solanopubamine (SPB) was evaluated for its antiviral activity against the hepatitis B virus [67]. The authors of this paper assayed the anti-hepatitis B function of Solanopubamine by ELISA of HBeAg and HBsAg, which are both serological markers of HBV DNA replication. These markers were evaluated in cellulo in HepG2.2.15 cells over 5 days. SPB exhibited significant inhibition of HBsAg and HBeAg after 3 days using 15–60 µM doses whilst having no cytotoxic effects on HepG2 cells at the maximal dose of 60 µM after 72h. SPB also showed anti-apoptotic effects on HepG2 cells and was linked to inhibition of caspase-3/7 in vitro in a DCFH-induced cell death assay, showing significant results at 30 and 60 µM. This antiviral effect was hypothesized in silico to be linked to two different synergistic factors via modeling and docking experiments: binding and inhibition of the HBV polymerase and inhibition of the caspase-3/7, pro-apoptotic proteins. Both factors could act to reduce infection and hepatotoxic properties of HBV.

The antiviral activity of tomatidine (TO) against dengue viruses (DENV), an arbovirus causing about 96 million symptomatic cases annually was reported by Diosa-Toro and co-workers [35]. Four serotypes (DENV 1–4) have been reported and all of them are responsible for the severe phenotype of the disease. In an MTT assay, TO showed no cellular toxicity at 10 µM towards Huh7 cells (CC_50_ = 80 µM), which are cells commonly used in antiviral studies. The highest selectivity index was found against the DENV-2 serotype (SI ~ 100; EC_50_ = 0.82 µM), following infection at a multiplicity of infection (MOI) of 1. Nevertheless, the EC_50_ values remained in the micromolar range (<5 µM) for all other serotypes. However, TO was less effective against other related flaviviruses (e.g., ZIKV) or even non-effective against others (e.g., WNV). TO reduced significantly DENV-2 infectivity when added pre-, during and up to 12 h post-infection (hpi) but no antiviral effect was observed after 16 and 20 hpi, showing the importance of timing in the administration of this antiviral. Moreover, the number of DENV-2 infected cells treated with TO during infection or 2 hpi was not statistically different and antiviral activity was still measured till 12 hpi. These results strongly support that TO does not interfere with the virus-cell binding, internalization or membrane fusion but rather acts at a stage after cell entry. On the other hand, TO likely acts before secretion of progeny virions, observed after 18 hpi, a timepoint when TO treatment was ineffective. TO was reported by Ebert et al. [36] to inhibit the expression of genes induced by the activating transcription factor 4 (ATF4), that is upregulated upon DENV infection [37]. The decrease in DENV infection upon TO treatment could partially be explained by inhibition of ATF4 expression. However, silencing of ATF4 reduced 50% of the production of infectious virus particles while a 99% reduction for TO-treated cells was observed. Accordingly, the effect of TO on ATF4 expression does not fully explain the potent antiviral activity of TO against dengue viruses. 

Later, the same group further investigated the in vitro antiviral activity of TO and structural analogs towards chikungunya virus [38]. TO (10 µM) was further validated as a non-toxic concentration on human cell line Huh7 (CC_50_ = 156 µM) using ATPLite assay. At this dose, it significantly reduced the production of progeny infectious virus particles (MOI 1) and is characterized by an EC_50_ value of 1.3 µM (SI = 120) against CHIKV-La Reunion strain (CHIKV East/Central/South African genotype). The antiviral activity of TO was also demonstrated against other lineages with a selectivity index of 41 (EC_50_ = 3.8 µM) and 78 (EC_50_ = 2.0) against the African S27 strain and Asian genotype (99659), respectively. Noteworthy, the antiviral effect of TO against CHIKV was also demonstrated in two other human cell lines (HFF-1 and U2OS), thus confirming the interesting properties of TO. Time-dependent administration studies showed that TO can reduce infectious particle production when added post-infection, with the strongest effect observed 2 hpi. However, unlike for DENV, no antiviral effect was observed when TO was administered pre- or during the infection. The authors hypothesized that the shorter replication time of CHIKV (8h compared to DENV’s 24 h) may not give sufficient time for TO to be internalized and to exert its effect. TO showed minor impact on the expression level of CHIKV E2-protein and likely interferes with multiple processes in the replicative cycle of CHIKV. It aborts viral infection after virus-cell binding and membrane fusion but before E2-protein translation and transportation to the cell-surface. The antiviral activity of TO was sustained for 24 hpi, maintaining a significant reduction of the production of progeny virus particles. It suggests that TO is capable of controlling CHIKV replication during several rounds of replication, showing interesting time-dependant properties. It is worth mentioning that the steroidal alkaloid solasodine also showed antiviral activity against CHIKV, albeit less potent than TO. 

Further studies have been conducted by Troost-Kind and co-workers to elucidate how TO interferes with CHIKV infectivity [39]. Using mass spectroscopy and western blot analysis, TO induced minor proteomic changes in Huh7 cells, including up-regulation of three cellular proteins (CD98, p62, metallothionein) and down-regulation of the thioredoxin-related transmembrane protein 2 (TMX2). However, TO antiviral activity sustained upon efficient knockdown of p62 and CD98 proteins, indicating that they are not involved in the antiviral process. In addition, TO does not seem to target CHKV’s entry since its antiviral potency was maintained in a virus entry-bypass assay. TO significantly reduced the level of intracellular viral RNA copy in infected cells (6 and 8 hpi) but once the replication is ongoing TO can no longer influence the process. However, TO strongly impacted the expression level of CHIKV proteins even during the replication. CHIKV nsP2, E1 and capsid protein expression was significantly reduced during infection, 4 hpi and 6 hpi. Accordingly, the authors proposed a mechanism for TO antiviral activity that relies on the control of viral protein expression early in infection, thus impeding efficient RNA replication and reducing the probabilities of infecting a cell. Further studies are still required to fully understand the role of cellular host protein in the antiviral activity of TO. Noteworthy, viral resistance could not be detected for CHIKV under selection pressure in the presence of TO. 

Recently, TO also demonstrated capabilities to significantly inhibit replication of the porcine epidemic diarrhea virus (PEDV) in vitro [40]. From a library of 911 natural products, Wang et al. identified TO (IC_50_ = 3.45 µM) as potent inhibitor of PEDV infectivity while showing low cytotoxicity (CC_50_ 45.7 µM) in Vero cells. Again, since TO-treatment only showed efficacy when added during the replication stage, TO inhibits PEDV infection by hampering viral replication. Using in silico molecular dynamics, the 3CL (nsp5) protease was identified as one of the potential replicative enzymes targeted by tomatidine. The interaction between TO and the enzyme 3CLpro was demonstrated by a fluorescence quenching assay while the binding affinity (Kd = 2.78 µM) was measured by isothermal titration calorimetry (ITC). Furthermore, TO inhibited the activity of PEDV 3CL protease in a dose-dependent manner, in a fragment cleavage visualization and FRET assay. The inhibition of PEDV 3CL protease by TO was about 47% and 75% at 25 µM and 50 µM, respectively. Interestingly, TO showed in vitro antiviral activity against other swine disease viruses including coronavirus (TGEV), arterivirus (PRRSV) and picornaviruses (EMC, SVA). Although, these viruses also contain 3C or 3CL protease, molecular docking analysis showed weak binding energy between TO and the viral 3C/3CL proteases of these viruses, thus suggesting that the antiviral activity of TO likely targets also other systems. 

Inspired by this study, Zrieq and co-workers performed molecular docking and dynamic simulation of TO to demonstrate its potential anti-SARS-CoV-2 effect through the inhibition of different enzymes including viral proteases 3CL, PL and endoribonuclease NSP15 [41]. These enzymes play a pivotal role in viral replication and transcription processes and were identified as potential therapeutic targets against SARS-CoV-2. In addition, viral infection induces a strong host inflammatory response and the activation of pro-inflammatory mediators such as COX-2 and phospholipase A2 (PLA2), which were thus highlighted as druggable targets to fight the inflammation process. Docking analysis demonstrated tight binding between TO and the selected targets (3CLpro, Plpro, NSP15, COX-2, PLA2). The protein-ligand interactions essentially occurred with an important catalytic residue of the enzymes, thereby highlighting the potential for SARS-CoV-2 treatment. The stability of the TO-protein complexes has been demonstrated through molecular dynamic simulation studies performed during 100 ns. TO established strong interactions with the surrounding amino acid residues of the active site of SARS-CoV-2 3Clpro, NSP15 and human COX-2. Besides, the mean binding free energy (ΔG_Bind_) for protein-ligand complexes was determined by post-dynamic MM-GBSA analysis and showed promising values (−47.46; −51.81; −54.89 Kcal.mol^−1^ for TO-3Clpro; TO-NSP15; TO-human COX-2 complexes, respectively). Remarkably, TO displays promising predicted drug-like properties, bioavailability, pharmacokinetics, and toxicity parameters using SwissADME web tool and PkCSM server, highlighting its potential as a new therapeutic agent. 

Noteworthy, the efficacy of TO to target PEDV-3CL protease and the structurally similar main protease (M^pro^ or 3Clpro) of SARS-CoV-2 was recently questioned by Vergoten et al. [42]. In silico molecular docking of available crystal structure of proteases (4XFQ and 5R80) were used to map the protein-ligand interactions. The TO-PEDV-3Clpro complex is mainly stabilized by hydrophobic interactions and a single hydrogen bond between the protein and the C-3 hydroxyl group of TO. Molecular models were constructed to determine the empirical energy of interaction (ΔE) and the free energy of hydration (ΔG) for the interaction of ligands with the two viral proteases. The calculated ΔE values (−45 Kcal.mol^−1^) and ΔG values (−20 Kcal.mol^−1^) obtained for TO-complexes are higher than the values for other ligands such as pneumocandin B0 (PNB01), thus indicating the enhanced potency of PNB01 on both proteases. These results suggests that TO likely acts as a multi-target agent, whereas more research is needed to fully elucidate the antiviral mode of action of TO against SARS-CoV-2.

*Solanum trilobatum* Linn. has been the target of a study performed by Anandakumar and coworkers targeting the M^pro^ 3CL protease of SARS-CoV-2 [70]. The authors identified via in silico screening three secondary metabolites of *S. trilobatum* which were able to bind to M^pro^, namely solanidine, solasodine and α-solanine, with binding energies of −69, −102 and −203 Kcal/mol, respectively. Although no direct proof of SARS-CoV-2 inhibition has been shown, preADMET toxicity prediction shows mutagenicity of solanidine in the AMES test, carcinogenicity of solanidine and solasodine in rats and ambiguous hERG activation for α-solanine, highlighting the toxic potential of solanum SGAs. In addition, the predicted activities of these S(G)As were assessed using PASS online (Prediction of Activity Spectra for Substances) and show probable CYP induction and inhibition of several cholesterol/hormonal inhibitions. However, solanidine and solasodine have been predicted to have acceptable PK-ADME parameters, even being predicted as having high gastrointestinal absorption.

### 5.5. Anti-Inflammatory

Anti-inflammatory behaviors of SGAs have been extensively researched in the last 10 years, including a review on steroidal saponin such as α-chaconine, α-solanine and lycoperoside H [174].

Solamargine (SM, *Solanum undatum*), a triglycoside derivative of solasodine, has shown promising anti-inflammatory properties, especially in UVB-induced inflammation models [82]. SM at doses of 1 µg/mL managed to reduce mRNA and protein expression of several pro-inflammatory markers in HaCaT cells, namely Il-1α, IL-1β, IFNγ and IL-8. Moreover, SM also diminished the expression of melanogenic factors α-MSH, ET-1 and bFGF, on both mRNA and protein levels. These results were also correlated with UVB-induced melanogenesis in cellulo by observing the melanosome maturation by examining the expression of critical melanogenic factors mRNA and protein levels. UVB induced the expression of melanogenic TYR, MITG, TRP-1 and TRP-2, while SM incubation managed to decrease those levels closer to the non-irradiated control. The authors also showed that the activity of SM was linked to the p38 MAPK/Nrf2 pathway, indicating that SM could activate the p38 MAPK signaling pathway after UVB irradiation.

Similarly to solamargine, khasianine (*Solanum nigrum*) is a glycosylated version of solasodine which has very recently demonstrated anti-inflammatory effects in skin inflammation models [88]. Firstly, khasianine was applied topically (5 µg/g gel) on the ears of IMQ induced psoriasis mice. Compared to the control group, the treatment significantly reduced the epidermal thickness and infiltrated immune cells. This result was confirmed by fluorescence microscopy, showing higher levels of CD45, CD4 and F4/80 positive cells in the epidermis of mice ears after induction, which was reduced back to normal levels when khasianine was applied. Immunohistochemical staining of the skin epidermis also showed reduced levels of TNF-α, p65, p-p65, IL-17A and IL-33, which all point towards a reduced inflammatory phenotype. Similarly, khasianine inhibited the TNF- α induced NF-κB activation in cellulo in human keratinocytes, as confirmed by the fluorescence of NF-κB p65 before and after treatment with khasianine. 

Solanine A, a steroidal alkaloid isolated from *Solanum nigrum Linn.* and characterized for the first time in 2018 by Gu et al., showed promising anti-inflammatory properties. It possesses a unique hexacyclic ring system (6/5/6/5/5/6) with an aldehyde functionality at C_7_ and significantly inhibits the nitric oxide release in LPS-induced RAW 264.7 macrophages (IC_50_ 3.85 0.71 µM) [175]. The same year, Zhao and associates further investigated the anti-inflammatory activity of solanine A [78]. The metabolite was shown to be completely non-cytotoxic on RAW264.7 cells at doses up to 20 µM. Moreover, this molecule showed significant inhibition on nitric oxide (NO) production and prostaglandin E_2_ (PGE_2_) at doses of 2.5 and 10 µM respectively. This effect was rapid, as even very short incubation times (<30 min) lead to significant drops in NO production in RAW264.7 cells. Solanine A did not only inhibit NO production but also drastically reduced the quantity of pro-inflammatory cytokines and chemokines mRNAs, namely TNF-α, IL-1β, IL-6 and CXCL9 at doses as low as 2.5 µM when activated by LPS/IFNγ. Lower amounts of pro-inflammatory proteins iNOS and COX2 were also observed at similar doses in cellulo. Solanine A also managed to inhibit LPS/INFγ stimulated ERK1/2 signaling in RAW264.7 macrophages, indicating a possible contribution to its anti-inflammatory potency, which was observed by lower levels of STAT-1 and 3 phosphorylation in western blotting. Finally, Solanine A was characterized in vivo for its anti-inflammatory effects and successfully reproduced the suppression of production of NO and PGE_2_ in mice serum. mRNA levels of TNF-α, IL-1β, IL-6 and CXCL9 also were lowered in mice receiving solanine A intraperitoneally at the dose of 5 mg/kg. Moreover, it exhibited anti-inflammatory properties in acute inflammatory xylene-induced ear edema and carrageenan-induced paw edema models, which was confirmed by the lower cell infiltration level by HE staining.

α-solanine (*Solanum tuberosum L.*) showed anti-inflammatory properties in a recently published paper by Shin and associates [71]. The experiments conducted are very similar to those used to prove the anti-inflammatory potential of solanine A published by Zhao [78]. α-solanine managed to lower the levels of NO, PGE_2_, TNF-α, IL-6 but not IL-1β in RAW 264.7 macrophages after LPS induction. The inhibitory effect was observed at 0.5 µM of α-solanine, while IL-1β inhibition was not observed at doses up to 5 µM. α-solanine also showed NF-κB inhibition behavior in LPS stimulated RAW 264.7 at doses of 2.5 µM and higher. Finally, α-solanine was administered per os in a mouse septic model after 25 mg/kg LPS i.p. injection. The administration of α-solanine induced a reduction in the mRNA of iNOS, COX-2, IL-6 and TNF-α at the dose of 2.5 µM. The compound even managed to induce IL-1β downregulation, even though it showed no activity in macrophages alone. The septic mice had slightly enhanced survival rates compared to the non-treated controls, from complete death at 30 h to 40% survival after 10 mg/kg per os administration of α-solanine.

The anti-inflammatory properties of α-solanine were also investigated by Piana et al. in an acute skin inflammation model in mice [72]. The authors showed that upon topical application, α-solanine inhibited croton oil-induced ear edema in a dose-dependent manner with an ID_50_ value of 0.004 mg/ear and a maximum inhibition of 80% of the edema at the highest dose of 0.37 mg/ear. α-solanine inhibited the increase of myeloperoxidase activity (MPO), a known marker for inflammatory response, when induced by the topical application of croton oil with an ID_50_ value of 0.012 mg/ear and a maximum inhibition of 46%. Besides, histological assessment of ear tissue highlighted that α-solanine (0.37 mg/ear) significantly reduced the edema and the inflammatory cell infiltration in comparison with the group treated with croton-oil only.

A-Chaconine, another SGA extracted from *Solanum tuberosum L.*, also represents a potential molecule for anti-inflammatory treatment [69]. In this 2015 paper, the authors investigated the mechanism as well as the anti-inflammatory properties of α-chaconine in cellulo using RAW 264.7 macrophages as well as in an LPS-induced sepsis model in mice. The authors investigated the relative mRNA levels of iNOS and COX-2, both important proteins in the production of NO and PGE_2_. α-Chaconine, in a dose-dependent manner, managed to reduce the levels of both studied mRNAs. Similarly, α-chaconine, at doses ranging from 200 nM to 2 µM, managed to dose-dependently reduce IL-1β, IL-6 and TNFα protein quantities and mRNA levels. α-Chaconine was subsequently incubated alongside LPS-induced RAW 264.7 macrophages in order to investigate the activation of P38, ERK1/2 and JNK. Results show the dose-dependent reduction in p-JNK via western blotting. This result was hypothesized as being a downstream effect of either MKK4 or MKK7 which could be hypophosphorylated. Results show downregulated p-MKK7, while MKK4 showed constant phosphorylation levels. JNK was finally downregulated using an siRNA, which completely abolished the previously described α-chaconine’s effect on IL-6 and TNF-α. Finally, the mRNA and protein quantities previously described were also evaluated in α-chaconine treated septic mice induced by LPS. All results were consistent with the in vitro experiments, but higher concentrations of α-chaconine were needed to elicit the response. Mice survival was also better, with an approximate 40% survival rate, compared to 0% for the control.

Recently, (25R)-22αN-4-nor-spirosol-5(6)-en-3β-ol-6-al-3-O-α-Lrhamnopyranosyl-(1→2)-[α-L-rhamnopyranosyl-(1→4)]-β-Dglucopyranoside, a structurally unique SGA isolated from *Solanum nigrum* L., has shown interesting properties on RAW 264.7 macrophages which are linked to anti-inflammatory behaviors [92]. The molecule showed an IC_50_ of 23.42 µM in an LPS-induced model of NO production, which is twice as low as the reference Indomethacin, highlighting the anti-inflammatory potential of these scaffolds.

In similar RAW 264.7 LPS-induced inflammatory assays, Abutiloside U (*Solanum melongena* L.) had an IC_50_ of 52.8 µM on NO production [93]. Several other non-alkaloid saponins in the family of Abutilosides also have anti-inflammatory effects, which indicates that the basic amine is probably not necessary for activity.

Lycoperoside H (LH) and its anti-inflammatory properties were investigated thoroughly by Takeda and associates using an IL-33Tg mice model which strongly expresses IL-33 in the epidermis, leading to inflammatory epidermis hyperplasia [97]. This model is characterized by a severe scratching behavior following the progression of the symptoms. LH-containing water was given once a day to these mice for 101 days at the dose of 10 mg/kg. This LH-containing diet was able to suppress the inflammation score of IL-33Tg mice, showing reduced skin inflammation, edema, bleeding, and soreness. TEWL (Transepidermal water loss) was also reduced in treated subjects. These results correlate with hematoxylin-eosin staining of the mouse facial skin, showing decreased epidermis thickness, eosinophil and mast cell invasion. Finally, ELISA analysis showed no significant changes in the quantities of IL-2, 4, 5 and IFN-γ in the skin, but showed significantly more IFN-γ in the blood of the IL-33Tg mice post-treatment. Total IgE quantities were also lowered in the serum, indicating a suppressed immune response to the inflammatory model compared to the control.

### 5.6. Osteoarthritis

Tomatidine (TO) has been proven as an antiarthritic molecule by Yu and associates in 2021 [46]. Firstly, TO diminished the viability of arthritic fibroblast-like synoviocytes (FLS) cells in the MTS assay, with an IC_50_ of 34 µM. This result was also confirmed in an EdU incorporation assay, showing a significant reduction in EdU incorporation at the dose of 5 µM of TO. Migration after 24h of TNFα-induced arthritic FLS cells was also reduced at the dose of 2.5 up to 10 µM of TO. TO incubation in cell cultures of TNFα-induced arthritic FLS also showed significant reduction in the quantity of mRNA of IL-1β, IL-6, TNFα, MMP-9 and RANKL at minimal doses of 2.5–5 µM. Likewise, IL-1β, IL-6 and TNFα protein quantification showed a significant reduction of the quantity of proteins at doses of 2.5, 5 and 10 µM of TO for each respective protein. TO also diminished synovial inflammation in a Collagen Induced Arthritis model in rats. Paw edema following induction significantly decreased using i.p. injections of TO at the doses of 5 and 15 mg/kg. Similarly, bone destruction in type II collagen-induced arthritis (CIA) rats was also significantly lower using the previously stated doses of TO. Interestingly, in vivo administration of TO in CIA rats significantly halted the body weight loss induced by the model at doses of 5 and 15 mg/kg i.p. Finally, TO suppressed the TNFα-induced phosphorylation of ERK. TO (10 µM) was able to block the translocation of NF-κB from the cytoplasm, as evidenced by confocal microscopy.

Osteoarthritis was investigated by Chu and associates in 2020 and in 2022 published an erratum, correcting a mistake in a figure [47]. TO has shown interesting properties, suppressing IL-1β-induced degradation of aggrecan and collagen-II, two biomolecules important for cartilage tissue wellness. These protein quantities were also rescued using tomatidine in a dose-dependent fashion over the doses of 2.5 to 10 µM. These results were confirmed in western blot experiments showing a dose-dependent diminution of IL-1β-induced upregulation of several matrix metalloproteinases, namely MMP1, 3 and 13 and ADAMTS-5 which are linked to the degradation of cartilage. Similarly, TO treatment suppressed the IL-1β-induced NF-κB pathway activation in the primary chondrocytes. Finally, rats treated with *per os* TO (25 mg/kg/day) for 12 weeks showed a reduction in the progression of the osteoarthritis in a rat model as a reduction in the Osteoarthritis Research Society International (OARSI) scores, INOS positive cells and MMP3 positive cells were diminished. 

### 5.7. Osteoporosis

A paper recently published by Hu and associates [44] showed the interesting activity of tomatidine (TO) in osteoclastogenesis, which could mitigate estrogen deficiency-induced bone mass loss (osteoporosis). Firstly, the authors showed that TO was non-toxic on cultures of BMM’s (Bone Marrow-derived Macrophages) up to 10 µM for 72 h. However, 100 µM of TO proved to be toxic, as only 25% of cells were viable after 72 h. BMMs were incubated with RANKL, the native ligand of the receptor activator of NF-κB (RANK), initiating the differentiation of BMM into osteoclasts. TO was shown to suppress the differentiation, by lowering the amount of TRAP+ cells in a dose-dependent manner over 2–8 µM. When TO was used early in osteoclastogenesis, it managed to decrease not only the number but also the size of BMMs, whereas late administration did not decrease the number nor the size of BMMs. TO treatment on RANKL stimulated BMMs also decreased the formation of F-actin rings as it is a marker of osteoclast function. Similarly, TO did significantly reduce the bone mass loss in a model of estrogen deficiency-induced osteoporosis in C57BL/6 mice as confirmed by micro-CT-scans. Tomatidine treatment similarly prevented the decrease of the maximum loading force on mice femurs following 8 weeks post-ovariectomy. TO’s role in osteoclastogenesis was also specified by qPCR, in which the authors observed a decrease in the number of mRNAs linked to osteoclast markers, namely NFATc1, CTR, DC-STAMP, cathepsin K, TRAP, c-Fos, MMP-9 and others. Finally, TO was shown to reduce the phosphorylation levels of JNK and P38 in RANKL-induced RAW264.7 cells. These results suggest that TO modulates bone resorption by mediating the activation of JNK/p38 MAPK pathway.

Later, Yu and co-workers further investigated the mechanism of action of TO to alleviate osteoporosis [45]. Using bioinformatics, the top-five shared KEGG (Krypto Encyclopedia of Genes and Genomes) pathways of TO-targeted genes and osteoporosis have been identified and all of them involve the p53 and/or MAPK signaling pathways. Remarkably, the p53 expression levels were significantly increased in the serum of osteoporosis patients compared to healthy individuals. On the other hand, in hMSC cells, SiRNA-p53 TO treatment significantly increased the expression of osteogenesis-related genes (ALP, OCN, Runx2) and the expression of cyclin D1/3 and Bcl2, whereas Bax expression was downregulated. Accordingly, the anti-apoptotic properties of TO by inhibiting the expression of p53 may likely promote osteoblast proliferation and inhibit osteoporosis. 

### 5.8. Skeletal Muscle Atrophy

Skeletal muscle atrophy is common as several conditions (bed rest, casting, malnutrition) and diseases (amyotrophic lateral sclerosis, osteoarthritis) might affect muscle mass. TO was screened by Dyle and associates [51,176,177] as a potential muscle atrophy inhibitor by a novel unbiased system-based method, showing high association between TO and muscle atrophy in PC3, HL60 and MCF7 cell lines. Accordingly, TO (1 µM) was firstly investigated for its hypertrophic properties on cultured C2C12 mice skeletal myotubes. Results show an overexpression of several anabolic factors such as enhanced protein levels, higher mitochondrial DNA levels and cell growth at the 24-h timepoint. Moreover, skeletal myotubes were shown to be of wider diameter after 24 h with doses as low as 300 nM of tomatidine. To follow up on the previous encouraging in cellulo results, TO was supplemented in standard chow (0.05%) of mice and the effects of this supplementation were observed five weeks later. The treated group showed higher phosphorylation of S6K, a marker of mTORC1 pathway activity. In relation to the previous in cellulo results, treated mice had enhanced protein quantities in the gastrocnemius, quadriceps and total cellular protein. Similarly, mitochondrial DNA was also significantly higher in the soleus and the gastrocnemius. TO doped chow created mice with greater muscular mass compared to the control, with about 13.7% more muscle mass. This increase was observed in the tibialis anterior, gastrocnemius, quadriceps, and triceps muscles. In older mice (~61 weeks-old), TO addition in standard chow also improved the grip strength, specific force and running distance after 9 weeks of treatment. Interestingly, TO also improved the body composition of mice after only 5 weeks of TO-doped chow while having no effect on total body weight. In fact, TO treatment was correlated with reduction in the epididymal, retro-peritoneal and scapular fat pads weight as well as reducing the diameter of adipocytes in H&E staining assays. Finally, the authors showed the effect of TO in intraperitoneal injections in a model of fasting-induced muscle atrophy. TO injection not only enhanced muscular mass of several muscle groups in mice, but also managed to rescue the muscle atrophy created by fasting, with levels similar to non-fasted mice. These results were also reproduced in a unilateral hind limb immobilization assay, in which TO lowered the muscular mass lost during the assay.

TO showed interesting results in an old mice model, as reported by Ebert et al. [36]. Tomatidine supplemented in standard chow (0.05%) was able to increase the muscular mass of 24-month-old mice as well as increase the diameter of Type IIb muscle fibers. This result was not only physiological, but also shown in enhanced grip strength and specific force of the treated mice with increases of 10 and 33%, respectively. Body weight was not affected by the TO treatment. Gene set enrichment identified several genes which were differentially expressed after TO supplementation. Most importantly, several encoding protein genes involved in muscle growth and mitochondrial bioenergetics were upregulated (Tfrc, Ckmt2) and ATF4, the protein of interest to this body of work, was reduced. The authors confirmed that ATF4 was involved in muscle synthesis, whereas the overexpression of this protein led to a lower quantity of muscle synthesis. In vivo, the knockout of this protein led to a rescue of muscle synthesis in aged ATF4 mKO mice. The previous in vivo results using a TO supplement were reproduced with the ATF4 mKO mice as they showed similar enhancement in skeletal muscle mass, grip strength and specific force. Remarkably, the inhibition of ATF4 by TO could also partially demonstrate the antiviral activity of TO against Dengue virus [37].

The tumor necrosis factor TNF-α is an inflammatory cytokine causing myoblasts apoptosis and playing an important role in skeletal muscle development eventually. A South-Korean group showed the involvement of tomatidine in apoptosis following TNF-α induction [52]. The authors showed that C_2_C_12_ myoblasts were susceptible to TNF-α induction, having greater quantities of cleaved caspase 3 and poly (ADP-ribose) polymerase (PARP), denoting an apoptotic phenotype. Reversely, incubation with TO was able to reverse the induction of TNF-α, reducing the quantities of the pro-apoptotic proteins at the dose of 10 µM. Flow cytometry also showed similar results. The authors clarified the role of TO, showing no involvement in the NF-κB signaling pathway, but showed a reduction of the p-JNK/JNK ratio, indicating a reduced apoptotic potential in this TNF-α induced model. 

### 5.9. Anti-Aging

Tomatidine (TO), the aglycone version of α-tomatine, was shown to extend lifespan and healthspan in *C. elegans* through mitophagy induction [53]. Under exposure of TO (25 µM), the lifespan of the nematode *C. elegans* was lengthened by about 7% whereas a higher concentration of TO (50 µM) showed toxicity issues with no significant changes in lifespan. TO administration (25 µM) also enhanced pharyngeal pumping and delayed the age-related decrease of swimming movement in *C. elegans*, thus indicating a reduction of physical decline due to aging. Analysis of muscle morphology strongly suggests that TO inhibits age-related sarcopenia in worms. Moreover, TO was shown to affect cell metabolism in worms that exhibit increased level of various free amino acids and oxidative phosphorylation. Gene expression analysis demonstrated that TO affects both oxidative mitochondrial function and ROS metabolism. It was shown that TO can influence mitochondrial cell quality and quantity by inducing mitochondrial mitophagy and biogenesis across species (*C. elegans*; rat cortical neurons; human cells). Mitophagy was revealed as pivotal to explain the health benefits of TO towards enhanced lifespan. Indeed, the worm mutants dct-1 and *pink-1*, deficient for two proteins pro executors of mitophagy, were no longer sensitive to TO benefits under the stress of the rotenone (mitochondrial complex I inhibitor). The authors eventually demonstrated that the anti-aging properties of TO rely on the activation of Nrf2/sKn-1 pathway and the activation of the mitochondrial unfolded protein response (UPR^mt^) that stimulate mitophagy.

Another study involving the lifespan of *C. elegans* has been published by Sarkar and associates in 2022 [54]. The authors identified that 1 mM of TO decreased α-synuclein aggregation and decreased general ROS levels. TO also managed to increase mitophagy-related genes *pink-1* and *pdr-1* by 1.42 and 2.39-fold, respectively, along with F25B4.7 and C05D11.9, two mitochondrial genes, which were increased by 145 and 189%. Finally, all these results lead to an increase in the maximal lifespan of *C. elegans* strain NL5901 from 20 days to 24 days.

### 5.10. Anti-Neurodegenerative

Several groups have investigated the potential of *Solanum* S(G)As in anti-neurodegenerative assays, especially in the context of Alzheimer’s. Researchers have identified the use of mitophagy inducers in order to reduce neuro-degenerative phenotypes [55].

A-Tomatine was evaluated for its neuroprotective properties as it is known to act as an antioxidative in cellulo. In a paper by Huang and associates in 2014 [65], α-tomatine was evaluated in hydrogen peroxide-induced cell death in SH-SY5Y cells, showing significant inhibition of cell death at the low-dose of 1 nM, with an effect similar to that of vitamin E (5 mg/mL) when incubated in 700 µM H_2_O_2_. α-Tomatine alone did not show any significant variability in cell viability assays. It also managed to reduce the amount of LDH released into the medium, post-exposure to 700 µM of hydrogen peroxide at the dose of 1 nM, in a dose-dependent manner. This result also indicates a lower amount of cell death than the non-treated control. As antioxidant properties are mediated by antioxidant enzymes, the activity of SOD, CAT, GPx and the content of GSH in SH-SY5Y were assayed. Hydrogen peroxide incubation on the cells induced a lower basal activity of all the previously listed enzymes and α-tomatine significantly rescued SOD and CAT at the dose of 1 nM, while significantly increasing GPx activity and GSH quantities at the higher 10 nM dose. α-Tomatine also significantly increased Bax protein expression in Western blotting at the dose of 1 nM, showing potential anti-apoptotic properties of tomatine, following H_2_O_2_ toxicity at 700 µM. This effect was also observed with pro-caspases-3/7 quantities which were also rescued using 10 nM of α-tomatine. The authors explain that increased pro-caspase quantities lead to lower quantities of caspase, indicating an anti-apoptotic phenotype; albeit no direct experiment has been conducted. Finally, α-tomatine also showed potent, dose-dependent BDNF protein rescue after hydrogen peroxide underexpression in the ranges of 1–100 nM. These results highlight the neurotropic effects of α-tomatine. 

The human toxicity of SAs/SGAs is often associated either to their complexation with membrane 3β-hydroxy sterols and disruption of the membrane, or to the inhibition of enzymes involved in the inactivation of acetylcholine (ACh), a naturally occurring neurotransmitter. High activity of such enzymes (e.g., acetylcholinesterase, AChE) is characteristic of pathologies such as Alzheimer’s disease and up to now, the cholinesterase antagonists donepezil, galantamine and rivastigmine are some of the only functional treatments for Alzheimer’s alongside the N-methyl-D-aspartate antagonist with memantine cotreatment. In this context, Taveira and associates further investigated the neuroprotective effect of α-tomatine and its aglycone TO against glutamate-induced toxicity in SH-SY5Y neuroblastoma cells [178]. First, the cholinesterase inhibition capacity of α-tomatine against both AChE (IC_25_ ~50 µM) and BChE (IC_25_ ~11 µM) was confirmed by colorimetric assay. Using MTT and LDH release assays, α-tomatine showed higher cytotoxicity (LC_50_ 4.64 µM) compared to TO (LC_50_ 457.12 µM) against SH-SY5Y cells. Nevertheless, α-tomatine (1.59 µM) and TO (126.25 µM) showed no cytotoxicity against various cell lines (SH-SY5Y, Caco-2, AGS). At such concentrations, α-tomatine increased by about 13% the LC_50_ value of glutamate in SH-SY5Y cells, while a minor effect was observed by the co-incubation of glutamate with TO. In a fluorescence assay, both metabolites proved capable of significantly decreasing reactive species and to preserve the mitochondrial potential compared to cells only exposed to glutamate. The role of nicotinic ACh receptors (nAChR) in the neuroprotective effects of α-tomatine and TO could be demonstrated. Indeed, when the cells were co-exposed to α-tomatine or TO with an AChR antagonist (mecamylamine) or α7-nAChR antagonist (MLA), the protective effect was abolished. In contrast, co-exposition with scopolamine, a muscarinic Ach receptor antagonist did not affect the neuroprotective properties of the steroidal (glyco)alkaloids.

Solanocapsine (*Solanum pseudocapsicum*) and several derivatives have also shown interesting antineurodegenerative properties as acetylcholinesterase inhibitors [95]. Solanocapsine showed an IC_50_ of 3.22 µM, while a synthetic compound, **24**, had an interesting 90 nM IC_50_ on AChE compared to the positive control Tacrine (Cognex). Most interestingly, **24** included in its structure the tricyclic amino quinoline scaffold of Tacrine, which created the most potent compound.

Recently, several analogs of solasodine, mainly with a seven-membered F-ring, were tested in biochemical assays against cholinesterases [90]. Firstly, solasodine did exhibit adequate binding affinities on AChE and BChE, with IC_50′_s of 28 and 6 µM, respectively. 22a(*N*)-Homosolasodine did exhibit slightly enhanced binding on both cholinesterases, with affinities of 8.5 and 7 µM. Curiously, the diastereoisomeric mixture of 22a(*N*)-Homosolasodine, with an unspecified configuration at C_22_ and C_25_ (F-ring), exhibits a unique selectivity towards BChE (>33-fold), with an affinity of 1.34 µM. Although this compound was the most potent one against both targets, the unspecified mixture at carbon 22 and the methyl of the F-ring of 22a(*N*)-Homosolasodine did exhibit a selectivity towards BChE of >33-fold, with an affinity of 1.34 µM. This compound was the only one capable of completely shifting the selectivity, as all the others were selective towards AChE. These compounds proved to be nontoxic at the dose of 10 µM on SH-SY5Y after 24 h. Most importantly, several of these molecules were able to rescue cell survival in a cell death model caused by glutamate. The most potent compounds (10 µM) managed to show 50% effective neuroprotection in this model of glutamate-induced cytotoxicity.

The neuroprotective properties of solasonine (SS) have been highlighted recently by Zhang et al. in a study where SS can relieve neurotoxicity induced by sevoflurane (SEV) [94]. In contrast with SEV exposure, SS treatment was shown to facilitate HT22 hippocampal neurons proliferation and to slow down apoptosis in a concentration–dependant manner. RT-qPCR and western blot experiments showed that SS treatment also increased the protein expression of several antioxidant factors (HO-1, NQO1, GCL, Prx1) dose-dependently. These results indicate that SS may reduce the SEV-induced oxidative stress and inflammation in neurons. Cell immunofluorescence and western blot assays were used to demonstrate that SS hinders the SEV-induced neurotoxicity by activating the FoxO3a signaling pathway and by inducing the expression of AMPK. These results were confirmed by showing that AMPK inhibition impairs the anti-apoptotic function of SS. SS treatment was able to decrease the levels of anti-inflammatory cytokines (TNF-α, IL-1β, HMGB1) and significantly up-regulated the antioxidant factors. However, AMPK inhibition abolished the positive effect of SS on inflammation in cellulo. In vivo, SS significantly enhanced the learning and memory capabilities of mice, when compared to non-treated SEV-exposed control group. Furthermore, SS (1 mg.kg^−1^) significantly up-regulated the AMPK/FoxO3a pathway in the hippocampus and improved damaged nerves of mice exposed to SEV. 

Kshirsagar and associates have investigated the impact of TO as a mitophagy inducer against phosphorylated Tau-induced mitochondrial and synaptic toxicities specifically in the context of Alzheimer’s disease [56]. TO (1 µM) was able to significantly reduce the appearance of pTau in HT22 cells transfected with Tau cDNA (named Tau-HT22 in this work). The authors showed enhanced mitochondrial dynamics by having greater fission protein concentrations (Drp1 and Fis1), while showing diminished fusion protein concentrations (Mfn2, Opa1) in Tau-HT22 cells incubated with TO. Similarly, biogenesis proteins were also upregulated following treatment (PGC1a, NRF1, 2 and TFAM). Mitophagy synaptic proteins were also upregulated, as shown by western blotting experiments (PINK1, Parkin, synaptophysin). Moreover, cell culture experiments showed a high survival rate of Tau-HT22 cells incubated with 10 µM of TO, displaying an increased survival from ~4% to ~20% in this severe phenotype. Interestingly, tomatidine treatment enhanced the maximal respiration and ATP production of these cells. Finally, electron microscopy experiments showed that tomatidine treatment reduced the number of mitochondria and observed a larger phenotype.

Kshirsagar published a second time a year later, this time using the amyloid precursor protein (APP) as a transfect for HT-22 cells (APP-HT22) [57]. Results are quite similar to the previously described article [56] albeit that TO (10µM) did not enhance the survival of APP-HT22 cells.

Wang et al. investigated the neuroprotective potential of TO after ischemic injury [58]. Firstly, the authors showed that TO at the dose of 10 µM did not elicit toxic effects against SH-SY5Y cells. In oxygen-glucose deprivation/reperfusion assays, TO (concentrations ranging from 1 to 10 µM) was able to rescue the survival of the cells post-starvation as seen in quantitative WST-1 absorbance and LDH release experiments. By western blotting, the authors also determined that the mitochondrial contents were unchanged as TOM20 and VDAC1 expression remained constant, independently from the TO concentration in the medium. 

Similarly, Ahsan and associates published neuroprotective properties of TO following ischemic injury, this time by investigating the lysosomal function in N2a cells [59]. Using the previously described oxygen-glucose deprivation/reperfusion assays (OGD/R), TO (3–10 µM) was able to rescue cell survival and lower LDH release in primary neurons and N2a cells. TO also managed to upregulate LC3-II in N2a cells following 4 h of depravation and 6h of reperfusion as assayed with chloroquine (CQ), which inhibits lysosomes and thus blocks the autophagic flux. Moreover, lysosome quantity and size were significantly greater in N2a cells after OGD/R when incubated with 10 µM of TO. Lysosomal protease activity assayed by DQ-BSA dye post-starvation was significantly lower but was rescued almost entirely after TO incubation. These results were explained partially by the overexpression of the two most common lysosomal proteases: cathepsin B and D which was 1.5 to 2 times more present when incubated with 10 µM of TO. TFEB also was more present in N2a cells post-incubation with TO after starvation. This protein is widely accepted as a master regulator for lysosome genesis, thus confirming the implication of TO in lysosomal activity.

### 5.11. Pulmonary Hypertension

An article published recently [73] linked pulmonary artery hypertension (PAH) treatment with the use of α-solanine. In Sugen5416/hypoxia-induced PAH mice, α-solanine managed to rescue partial pressure of oxygen (PaO_2_), cardiac output and mean right ventricular systolic pressure after two weeks of daily 5 µg/kg i.p. injections. This molecule did show dose-dependent growth inhibition of pulmonary artery smooth muscle cells (PAH-PASMC) and pulmonary artery endothelial cells (PAH-PAEC) with an IC_50_ of ~ 10 µM. α-solanine also dose-dependently decreased the proliferation of PASMC cells with doses up to 10 µM in proliferating cell nuclear antigen (PCNA) assay, ki67 staining and EdU incorporation assays. α-solanine also increased the sensitivity of PASMC to serum deprivation-induced apoptosis at doses of 50–100 µM. AXIN2 expression was also rescued as well as β-catenin using α-solanine, effectively reversing the bone morphogenetic protein receptor type-2 (BMPR2) downregulation in the PAH model. This result was also confirmed in calcium indication assays, in which PAH induced increases of Ca^2+^ and α-solanine reversed that amount back to normal. In PAH mice, induced with monocrotaline, α-solanine (5 mg/kg/day, i.p.) managed to reverse remodeling of distal pulmonary arteries, reduced the arterial hypertension, and reduced right ventricular hypertrophy. Similarly, it also reduced the proliferation of endothelial cells in a PAH model, while improving the sensitivity to apoptosis as measured by Annexin V. In a wound healing assay, α-solanine incubation (10 µM) accelerated the healing process. At the same dose, it also suppressed the activation of Akt signaling pathways in hypoxia-pulmonary artery endothelial cells. These results were observed by reversing the hypoxia-induced increase of phosphorylation of GSK-3α and Akt in western blot. The authors have even proposed a cascade scheme for α-solanine which involved various proteins (AXIN2, BMPR2, Akt, etc.) and their signaling pathways pointing toward reversal of the pulmonary vascular remodeling. Moreover, α-solanine treatment is known to increase caspase-3 protein level in cancer cells thus suggesting that the pulmonary-artery protective effect of α-solanine also relies on caspase-3 mediated PASMC apoptosis [74,75].

### 5.12. Asthma

Asthma treatment has been tackled by Kuo et al. in a recent article where the authors compare the effect of tomatidine (TO) with prednisolone, an approved steroid medication [60]. Asthmatic mice were subjected to methacholine, a bronchial contracting agent and were treated with TO or ovalbumin as a challenge group. TO-treated ovalbumin-challenged mice showed significantly reduced airway hyperresponsiveness at the doses of 0.5 to 5 mg/kg i.p., with effects similar to the prednisolone group. TO also suppressed the production of several pro-inflammatory chemokines and genes, namely IFN-γ, IL-4, 5, 13, iNOS, MUC5AC, and gob5 and neutrophil infiltration at various doses ranging from 0.5–5 mg/kg. The previously mentioned interleukins were also reduced in the serum of TO-treated rats.

From the same group, similar results were produced using TO as a treatment in inflammation following acute lung injury using a LPS-induced inflammatory model [61]. This research work also highlighted diminished neutrophil infiltration, diminished pro-oxidative and pro-inflammatory proteins and chemokines in the serum. Although the first results are very similar, the authors identified the NF-κB and MAPK pathways to be of importance as p65 phosphorylation was lowered in the treatment group.

Arora and associates showed the promising therapeutic properties of solasodine in an ovalbumin-induced bronchial asthma model [179]. As previously reported for other steroidal alkaloids in the anti-inflammatory section, solasodine treatment induced a significant diminution of pro-inflammatory mediators, including IgE, nitrites, nitric oxide, TNF-α, IL-1β, LTD-4, IL-4 and IL-5. Solasodine (1 mg/kg and 10 mg/kg) also managed to show in vivo activity, reversing OVA-induced airway hyperresponsiveness, reducing infiltration of inflammatory cells and histamine levels.

### 5.13. Antitussive

One study by Dan Li et al. reported the antitussive effects of spiralosides A–C, new steroidal alkaloids extracted from *Solanum spirale* [89]. The methanolic extract of the plant was evaporated and partitioned between EtOAc and 0.5% HCl to yield the “total extract” which was later shown to contain several spiralosides. The total extract was shown to reduce ammonia liquor-induced cough in mice with about 30% inhibition at the dose of 10 mg/kg p.o., while the gold standard, codeine phosphate, reduced cough symptoms by about 60% at the dose of 30 mg/kg p.o. Similarly, the protective expectorant activity was assayed using phenol red secretion test in mice. Total extract however showed no significant phenol red secretion compared to the control. The spiralosides A–C were also tested in cytotoxic assays against HL-60, A-549, SMMC-7721, MCF-7 and SW480 cell lines and showed low cytotoxicity with IC_50′_s of >40 µM for all compounds screened.

### 5.14. Cardioprotection

In a paper by Chowański et al. [9], α-chaconine and α-solanine were used in conjunction with verapamil in cardioinhibitory assays on *T. molitor* semi-isolated heart. Firstly, both SGAs exhibit cardioinhibitory behavior as 10^−7^ M of α-solanine shows significant changes in contraction frequency with an IC_50_ of 1.88 × 10^−5^ M while α-chaconine shows significant change in contraction frequency at 10^−6^ M, with an IC_50_ of 7.48 × 10^−5^ M. All in all, α-solanine had the most potent action, with its maximal action being reached at an earlier time than that of α-chaconine. Since the SGAs and verapamil both cause cardio-inhibition, the authors thought that co-incubation could cause an additive effect. When α-solanine was used in conjunction with verapamil (3 × 10^−5^ M), an antagonistic effect was observed, rather. Although the authors think that it acts as an antagonist, α-solanine could also act as a partial agonist competing with verapamil in the protein’s binding site. α-chaconine also created the same effect, but to a lower degree. This hypothesis was later confirmed by adding the SGAs after verapamil and observing the recovery time of cardiac frequency. Both α-chaconine and α-solanine increased the time for recovery, implying a cardioinhibitory effect.

Kim and associates have shown the role of TO in cardiac disease modeling by investigating its impact on embryonic stem cell-derived cardiomyocytes (hESC-CMs) [62]. Firstly, the authors showed that TO was able to induce α-sarcomeric actinin (α-SA) in hESC-CMs at the doses of 0.5 µM to 2 µM. Interestingly, quantities of 2 µM did not induce an expression as high as the 1 µM dose, probably because of the toxicity of TO. At 1 µM, it also induced the expression of cardiac structural proteins, mainly cardiac troponin C, myosin light chain isoforms 2,7 and myosin heavy chain isoform 7. Macroscopical changes in hESC-CMs were also observed post-incubation with tomatidine, showing signs of cardiac hypertrophy and sarcomere organization in cell culture. Similarly, T-tubules (essential for the establishment of excitation-contraction coupling) were also shown to be denser and in greater amounts in tomatidine incubated hESC-CMs. As mitochondrial function is often linked to cardiomyocyte maturation, the authors investigated the oxidative potential of the mitochondria of hESC-CMs and found higher maximal respiratory capacity and greater ATP production in TO-treated cells. Moreover, more mitochondria were present inside the cells when compared to the non-treated controls. Finally, the membrane potential of the mitochondria was assayed using an immunofluorescence assay with TMRM staining. Results show TMRM staining significantly higher in the TO treated cells, suggesting an enhanced membrane potential. Maturation of the hESC-CMs was also more rapid as electrophysiological assays show a more prolonged action potential duration, reduction in the spontaneous beating frequency and an increase in the spike amplitude, coupled with a superior conduction velocity.

### 5.15. Anti-Obesity

Tomatidine (TO) was recently highlighted for the development of new therapeutic agents to treat obesity and non-alcoholic fatty liver disease (NAFLD). Kusu and coworkers were the first to show that the steroidal alkaloid significantly supresses the lipid accumulation and lipotoxicity in palmitate-exposed HepG2 hepatocytes [63]. Immunoblotting assay demonstrated that TO causes activation of the AMP-activated protein kinase (AMPK), via ACC1 phosphorylation, in a dose dependent manner. It also modulates two pivotal transcription factors, FoxO1 and SREBP-1, involved in hepatic metabolism and homeostasis. Indeed, the cellular levels of FoxO1 and the expression of its transcriptional factor ATGL (fatty triglyceride lipase) were upregulated upon TO treatment, whereas lipogenesis was attenuated by downregulation of SREPB-1 and the expression of its transcriptional factor FAS (fatty acid synthase). Noteworthily, TO’s effect was significantly abolished either by co-treatment with a selective Ca^2+^-calmodulin-dependent protein kinase kinase β (CaMKKβ) or by the knockdown of the nuclear hormone receptor VDR (vitamin D receptor). Furthermore, using a fluorescent Ca^2+^-probe, TO proved to induce a dose-dependant increase of Ca^2+^ intracellular concentration. These results strongly suggest that TO acts as an agonist of VDR receptor and stimulates AMPK phosphorylation by CaMMKβ activation via Ca^2+^ signaling. The activated AMPK regulates key transcription factors of hepatic lipid metabolism (FoxO1 and SREBP-1), thus inhibiting lipid accumulation in HepG2 cells. Later, Wu et al. demonstrated the in vivo efficacy of tomatidine to reduce the body weight of obese mice fed with high-fat diet (HDF) [64]. TO treatment significantly decreased epididymal adipose tissue weight and liver steatosis in HDF-induced obese mice. Indeed, TO reduced not only the liver weight, but also the number of lipid droplets in liver tissue and the NAFLD score compared to non-treated obese mice. The serum levels of triglycerides, total cholesterol, alanine aminotransferase and aspartate aminotransferase were also diminished upon TO-treatment, while glycogen quantities were higher. Accordingly, TO affects serum lipid metabolism by increasing HDL levels and decreasing the free fatty acid in HFD-induced obese mice. The authors further confirmed in vivo in obese mice and in vitro (FL83B cells) that TO inhibits lipogenesis and activates lipolysis. Indeed, TO significantly reduced cellular levels of C/EBPβ and SREBP-1c and inhibited FAS expression, thus hampering lipid synthesis and improving liver steatosis in HFD-induced obese mice. On the other hand, TO was shown to downregulate the phosphorylation of ATGL (fatty triglyceride lipase) and HSL (hormone sensitive lipase) to accelerate lipolysis in the liver. The β-oxidation pathway of fatty acids, pivotal for the turnover of triglycerides, was also shown to be affected by TO treatment with an increase of the CPT-1 and PPAR-α expression in the liver tissue. In conclusion, TO reduces lipid accumulation in the liver tissue by enhancing the Sirtuin-1/AMPK pathway, by promoting lipolysis and β-oxidation and by impeding lipogenesis.

### 5.16. Anti-Embryonic

α-Solanine has been shown to have anti-embryonic properties in pig in vitro oocyte maturation models [76]. Lin and associates showed that α-solanine inhibited the growth of cumulus-oocyte complexes (COCs) incubated with 10–50 µM of the steroidal alkaloid. Moreover, α-solanine also induced morphological abnormalities and bigger diameters in oocytes at the same doses. Similarly, α-solanine was found to be highly toxic starting from 10 µM, inducing cell death in oocytes and COCs with less than 10% viability in both experiments. This toxic effect was linked to influences in the meiotic cycle in which oocytes were found to be stuck in the germinal vesicle breakdown and metaphase 1. Moreover, oocytes incubated with α-solanine showed higher percentages of abnormal spindle formation and abnormal cortical granule (CG) distribution, a marker of successful cytoplasmic maturation. α-Solanine (10 µM) treatment induced autophagy in pig oocytes, as seen by higher mRNA quantities of ATG3, ATG5, ATG7, LAMP2, LC3 and mTOR, which are autophagy-related genes. Similarly, apoptosis was also upregulated, showing higher levels of cells in apoptosis as assayed by the TUNEL experiment and higher levels of BAX and caspase-3. Finally, the developmental quality and capacity of oocytes was evaluated while incubated with α-solanine. Results show lower blastocysts levels, with less individual cells comprised in the blastocyst. Also, a TUNEL assay showed higher levels of apoptosis, which was also correlated with higher BAX mRNA quantities.

Chen and associates also reported similar results using α -solanine in the trophoblast of HTR-8/SVneo cells, which highlights the anti-embryonic properties of this metabolite [77]. Their results show an accumulation of cells which are stuck in S and G2 phases of replication, while having less cells in the G0/G1 phase at the dose of 30 µM of solanine. Similarly, 20–30 µM of solanine induced significant apoptosis and autophagy in HTR-8/SVneo cells and significantly reduced the migration and invasion of these cells in vitro.

### 5.17. Cancer Treatment

Cancer applications of SGAs have been extensively reviewed in the last 10 years [166,167,168,180,181,182,183,184,185,186,187].Accordingly, we decided not to dive deeply into this field of research. However, in the last 10 years, several SGAs have shown interesting anti-cancer properties, which are listed in Table 2.

### 5.18. Summary

Table 1 and Table 2 offer a unique overview of the pharmacological effect related to each *Solanum* steroidal (glycol)alkaloids over the last ten years. On the other hand, Figure 6 summarizes the pharmacological mechanisms and the biological response induced by the solanum SGAs. It intends to inspire future research to better understand the bioactivities of these natural products and to establish new structure-activity relationship.

## 6. Toxicity & Therapeutic Challenges

*Solanum* steroidal (glyco)alkaloids have been traditionally used in some cultures for medicinal purposes, including as a treatment for inflammation, pain, and fever. However, the use of these alkaloids as drugs is controversial due to their potential toxicity and side effects. While several studies suggest that *Solanum* steroidal (glyco)alkaloids may have therapeutic benefits, there is still limited clinical evidence to support their use as drugs. Furthermore, the toxicity of these alkaloids at moderate doses raises concerns about their safety, and regulatory agencies such as the FDA have not approved any *Solanum* steroidal alkaloids as drugs. Research on the toxic effects of *Solanum* SGAs has found that they can cause a range of adverse effects, including:

### 6.1. Gastrointestinal Distress

Consumption of high levels of *Solanum* steroidal alkaloids can lead to gastrointestinal symptoms such as nausea, vomiting, abdominal pain, and diarrhea. This is thought to be due to the irritant effects of the alkaloids on the gastrointestinal tract [188].

### 6.2. Neurological Symptoms

*Solanum* steroidal alkaloids can affect the central nervous system, leading to symptoms such as headache, dizziness, confusion, and even coma, probably due to the ability of the alkaloids to disrupt the normal functioning of neurons [189]. For example, solanine has been shown to have neurotoxic effects by inhibiting acetylcholinesterase, an enzyme involved in the breakdown of acetylcholine. This inhibition can lead to an accumulation of acetylcholine in the nervous system, which can cause symptoms such as dizziness, confusion, and hallucinations. Long-term exposure to α-solanine has been associated with the development of neurological disorders such as Parkinson’s disease and Alzheimer’s disease [190].

### 6.3. Cardiac Effects

Some *Solanum* SGAs have been found to have toxic effects on the heart, leading to arrhythmias and other cardiac disturbances. This is thought to be due to the ability of the alkaloids to interfere with ion channels in the heart [191]. Recently, α-solanine and α-chaconine were identified to display cardiomodulatory activity by interacting with the drug verapamil, that decrease the efficiency of cardiovascular therapy [9].

### 6.4. Respiratory Effects

In severe cases of *Solanum* steroidal alkaloid toxicity, respiratory failure can occur, leading to difficulty breathing, respiratory distress, and even death [192].

### 6.5. Renal Effects

*Solanum* steroidal alkaloids have been found to have toxic effects on the kidneys, leading to renal failure in severe cases [193].

### 6.6. Teratogenic Effects

Studies suggest that high-food consumption of “potato SGAs” α-solanine and α-chaconine during the periconceptional period may increase the risk of neural tube defects (NTDs) and orofacial clefts (OFCs) by inhibiting JNK in the Wnt/PCP pathway [194].

These toxic effects can occur even at relatively low doses of the alkaloids, making it difficult to establish a safe and effective therapeutic dosage. It is important to note that the toxic effects of *Solanum* steroidal alkaloids can vary depending on the specific alkaloid, the dose, and individual factors such as age, health status, and sensitivity. Another concern with the use of *Solanum* steroidal alkaloids as drugs is the lack of standardization in the preparation of these compounds. Different species of *Solanum* plants contain different types and concentrations of steroidal alkaloids, and even within a single species, the alkaloid content can vary depending on factors such as the time of year, growing conditions, and harvesting practices. This variability makes it difficult to ensure consistent and reliable dosing of *Solanum* steroidal alkaloids in drug preparations.

On the other hand, it is possible that a medicinal chemistry approach could improve the therapeutic benefits of *Solanum* steroidal (glyco)alkaloids. One approach that could be used is to modify the structure of the alkaloids to enhance their potency and selectivity for specific biological targets. For example, structural modifications could improve the alkaloids’ ability to bind to and activate specific receptors or increase their solubility and bioavailability. Another approach would be to identify and isolate specific *Solanum* compounds with their most promising therapeutic properties and then optimize their pharmacokinetic and pharmacodynamic properties. This could involve modifying the alkaloids’ chemical structure to improve their absorption, distribution, metabolism, and excretion, as well as their potency and selectivity. Any efforts to improve the therapeutic benefits of *Solanum* steroidal alkaloids would need to be balanced against the potential toxicity and side effects of these compounds. Overall, *Solanum* steroidal (glyco)alkaloids have a diverse range of biological activities and modes of action, some of which have potential therapeutic applications. However, their toxicity and side effects must also be carefully considered before their use as drugs or food additives. Further research is needed to better understand the safety and efficacy of *Solanum* steroidal alkaloids, and to develop standardized preparations that can be reliably used in clinical settings.

## 7. Conclusions

Steroid (glyco)alkaloids of the *Solanum* genus were associated with an impressive variety of 17 potential therapeutic applications over the last ten years (i.e., 2012–2022). Although the primary role of these metabolites is to protect the plant from its environment, this review highlights the structural diversity offered by the *Solanum* steroidal (glycol)alkaloid family and their great potential regarding new therapeutic development. Nevertheless, SGAs and their aglycones are multitarget agents, interfering with several signaling pathways and therefore have very complex interactions with the human body. Remarkably, SGAs and their respective steroidal alkaloids induce different pharmacological responses. Therefore, structural information about their interaction with molecular targets, the study of their mechanism of action as well as their pharmacokinetic profile are determinant to define the appropriate therapeutic window and overcome their known toxicological and anti-nutritive properties. To date, a huge gap still exists between the high therapeutic value of *Solanum* S(G)As and their clinical outcome, likely because of the lack of pharmacokinetic data and their poor physicochemical properties, thus limiting translation between in vitro and in vivo studies. In this context, further structural—activity relationship studies are needed to improve the drug-like properties, safety profile and potency of *Solanum* SGAs. This review offers a unique overview of the pharmacological effects induced by *Solanum* steroidal glycoalkaloids and may inspire upcoming research.

## Figures and Tables

**Figure 1 molecules-28-04957-f001:**
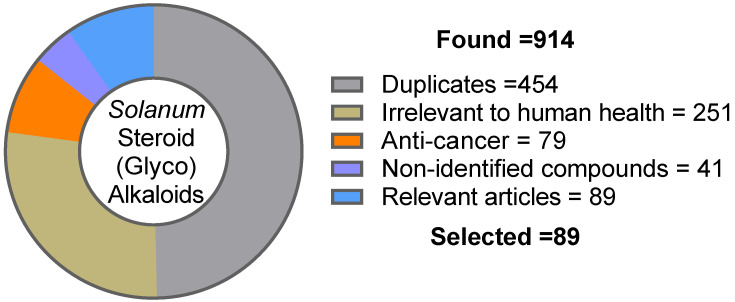
Diagram of the selection process for the scoping review.

**Figure 2 molecules-28-04957-f002:**
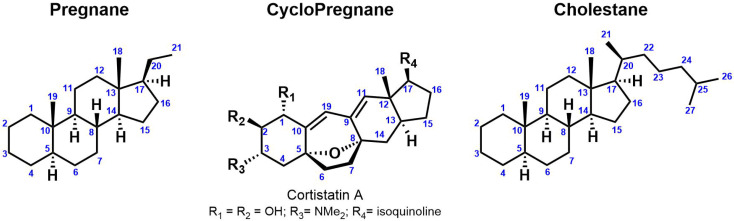
Steroid categories according to their carbon framework.

**Figure 3 molecules-28-04957-f003:**
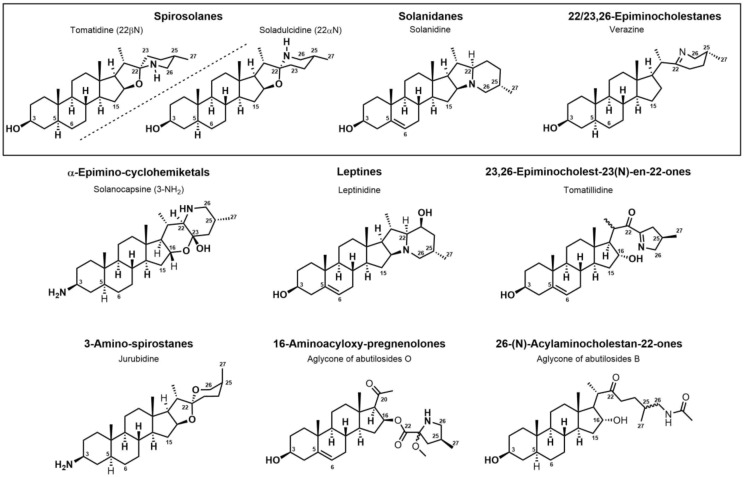
Classification of steroidal alkaloids.

**Figure 4 molecules-28-04957-f004:**
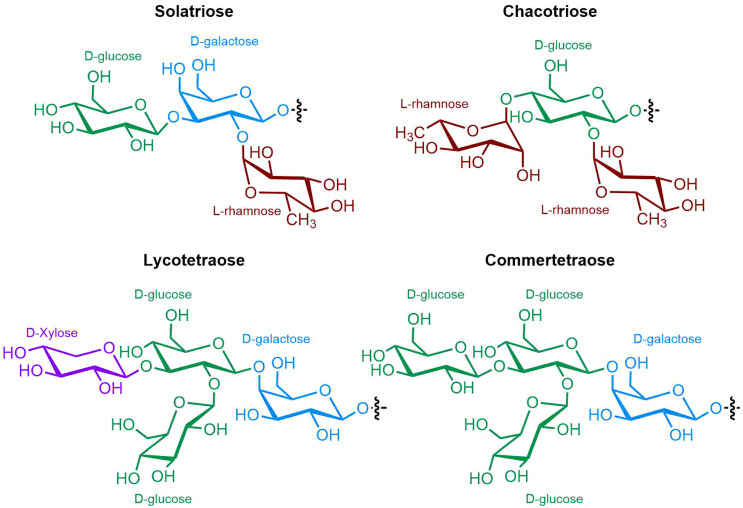
Chemical structures of the main oligosaccharide units of Solanum SGAs.

**Figure 5 molecules-28-04957-f005:**
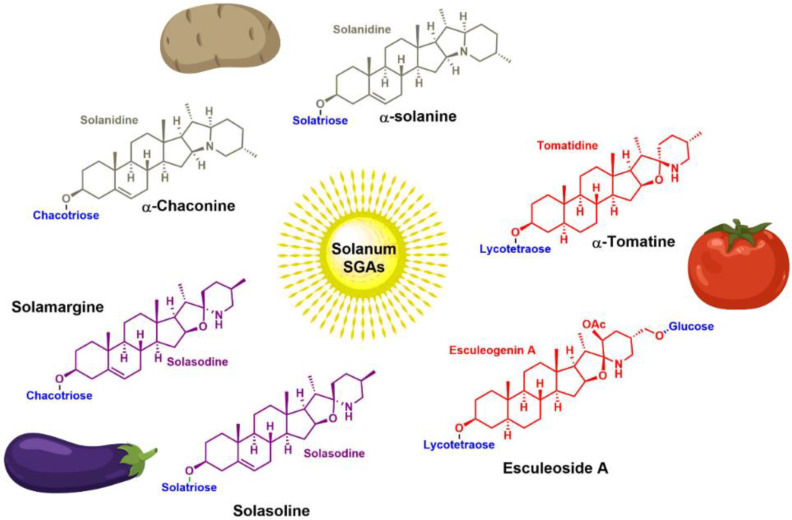
Chemical structure of some popular *Solanum* SGAs isolated from potato, tomato, and eggplant.

**Figure 6 molecules-28-04957-f006:**
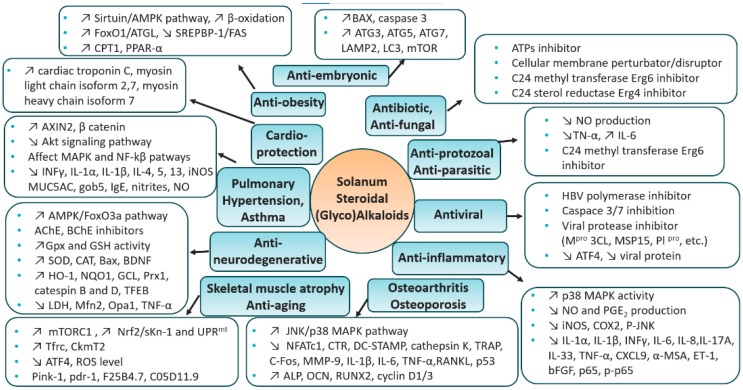
Summary of the pharmacological effects induced by *Solanum* SGAs. (Up arrow) = overexpressed, synthesis or activation. (Down arrow) = underexpressed, degradation or inactivation.

**Table 1 molecules-28-04957-t001:** SGAs and their respective non-cancer applications.

Compound	Structure	Applications Ref.
**Tomatidine**	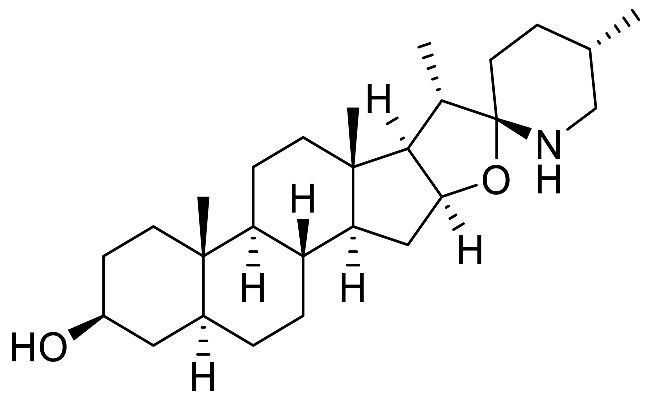	**Antibiotic or antibiotic adjuvant** (*S. aureus* [24,25,26,27,28,29,30,31,32], *L monocytogenes* [30,33], *B. subtilis* [30], *P. aeruginosa* [27,29], *E. faecalis* [27]) **Antifungal** (*C. krusei* [34], *C. tropicalis* [34], *C. parapsilosis* [34]) **Antiviral** (DENV-2 [35,36,37], CHIKV [38,39], PEDV [40], SARS-CoV-2 [41,42]) **Anti-leishmanial** (*L. amazonensis* [43]) **Osteoporosis** [44,45] **Osteoarthritis** [46,47,48] **Anti-protozoal** (*T. cruzei* Tulahuen C2C4 strain [49], *P. serpens* [50]) **Skeletal muscle atrophy** [36,51,52] **Anti-aging** [53,54] **Anti-neurodegenerative** [55,56,57,58,59] **Asthma** [60,61] **Cardio-protection** [62] **Obesity** [63,64]
**α-Tomatine**	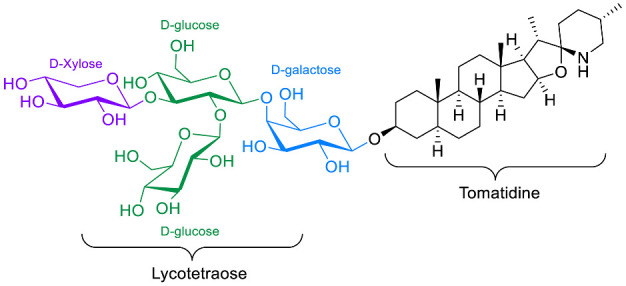	**Anti-neurodegenerative** [65] **Anti-protozoal** (*T. fetus feline, T. fetus bovine, T. vaginalis human)* [66]
**Solano-pubamine**	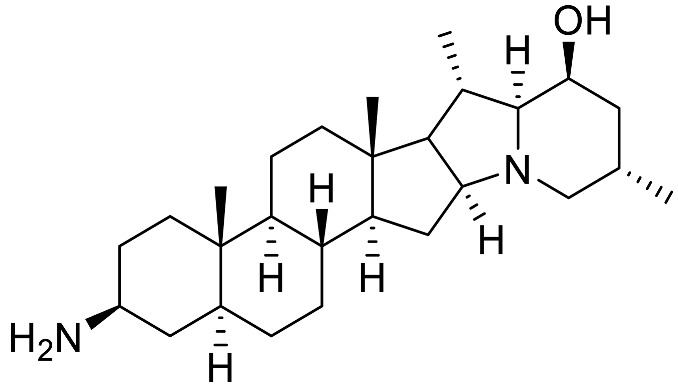	**Antiviral** (HBV [67]) **Antifungal** *(C. albicans*, *C. tenuis)* [68]
**α-Chaconine**	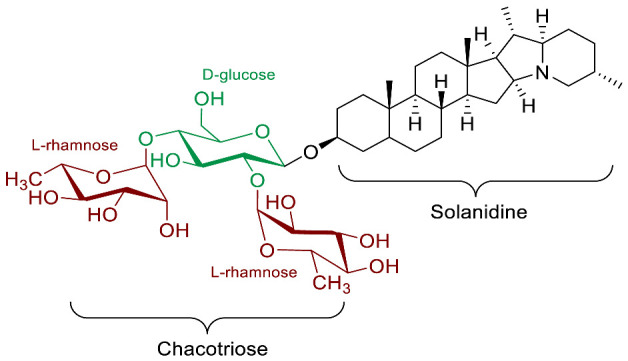	**Cardio-protection** [9] **Anti-inflammatory** [69] **Anti-protozoal** (*T. fetus feline, T. fetus bovine, T. vaginalis human)* [66]
**α-Solanine**	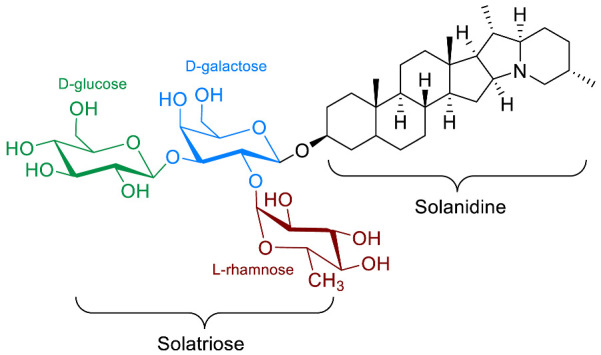	**Antiviral** (SARS-CoV-2 [70]) **Cardio-protection** [9] Anti-inflammatory [71,72] **Pulmonary hypertension** [73,74,75] **Anti-embryonic** [76,77] **Anti-protozoal** (*T. fetus feline, T. fetus bovine, T. vaginalis human)* [66]
**Solanine A**	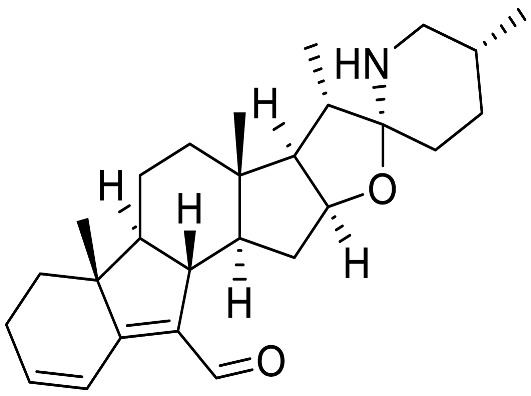	**Anti-inflammatory** [78]
**Solasodine-3-*O*-β-D-glucopyran-oside**	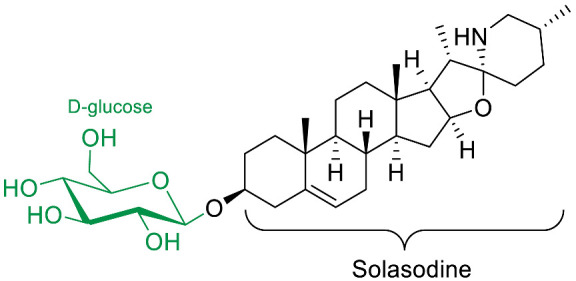	**Antifungal** (*C. albicans* [79,80])
**Solamargine**	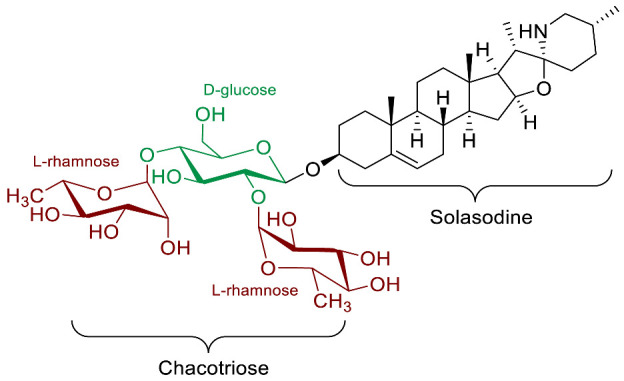	**Antibacterial effects** (*P. aeruginosa* [81]) **Anti-inflammatory** [82] **Anti-leishmanial** (*L. amazonensis* [83], *L. mexicana* [84]) **Antifungal** *(T. lentagrophytes* [81]) **Anti-protozoal** (*G. lamblia* [85], *T. cruzei* Y stain [86]) **Molluscicidal** *(B. alexandrina* [87]) **Schistosomicidal** (*S. mansoni* [87])
**Khasianine**	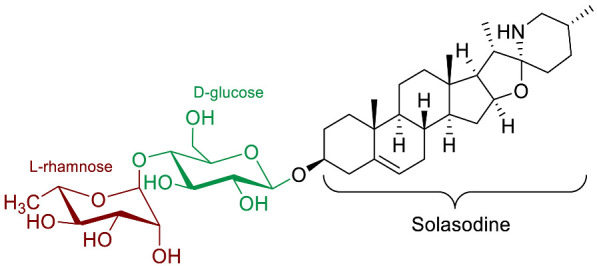	**Anti-inflammatory** [88]
**Spiralosides A-C**	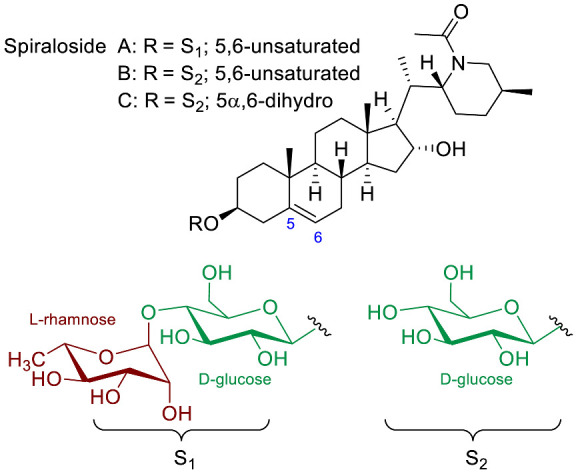	**Antitussive [89]**
**Solasodine +** 22a(*N*)-Homosolasodine	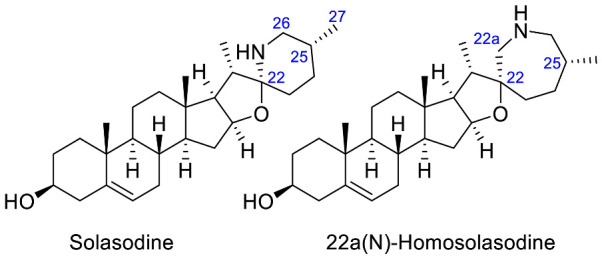	**Antineurodegenerative** (Alzheimer’s [90])
**Solanindin B**	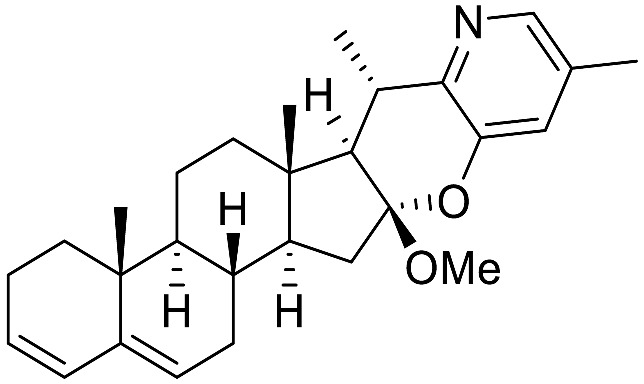	**Antibiotic** (*S. aureus* [91])
**(25R)-22αN-4-nor-spirosol-5(6)-en-3β-ol-6-al glycosides**	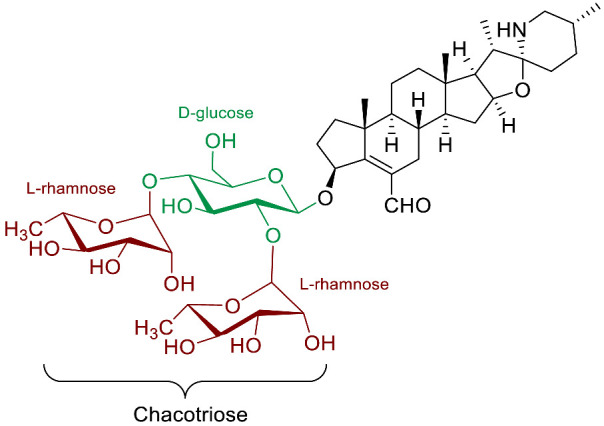	**Anti-inflammatory [92]**
**Abutiloside U**	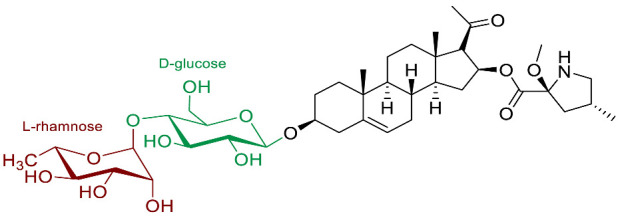	**Anti-inflammatory** [93]
**Solasonine**	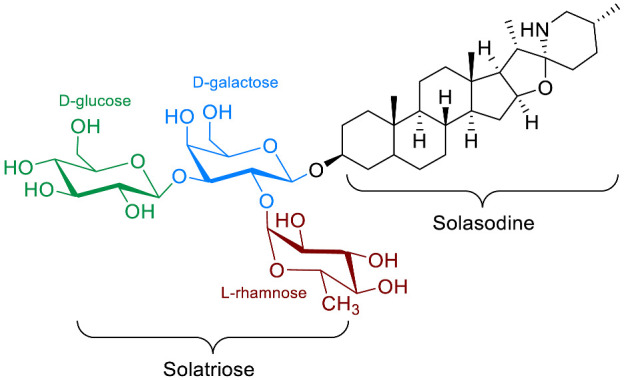	**Antiviral** (SARS-CoV-2 [70]) **Anti-leishmanial** (*L. amazonensis* [83], *L. mexicana* [84]) **Anti-protozoal** (*G. lamblia* [85]) **Molluscicidal** *(B. alexandrina* [87]) **Schistosomicidal** *(S. mansoni* [87]) **Anti-neurodegenerative** [94]
**Solanocapsine and derivatives**	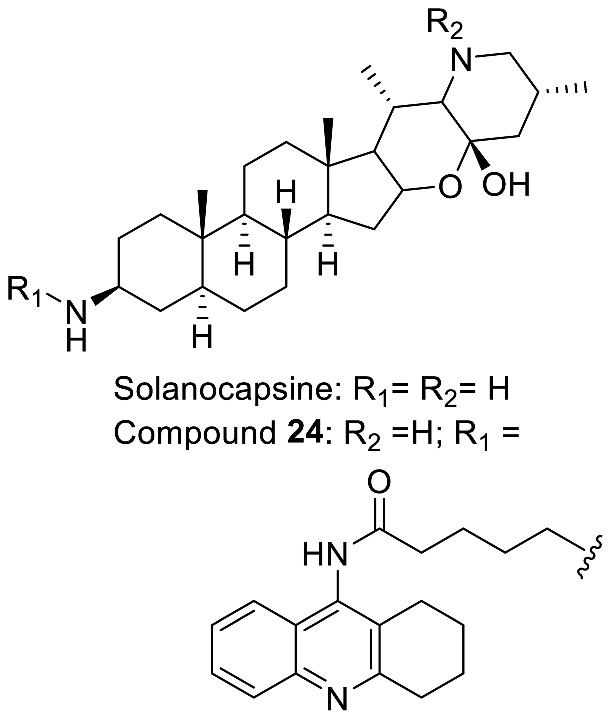	**Anti-neurodegenerative** [95]
**β-Solamarine**	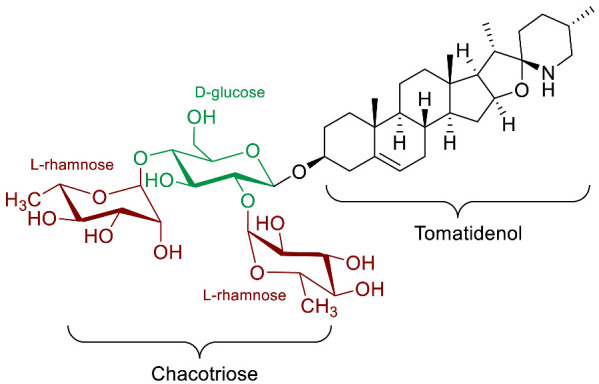	**Molluscicidal***(G. truncatula Müll*. [96])
**Lycoperoside H**	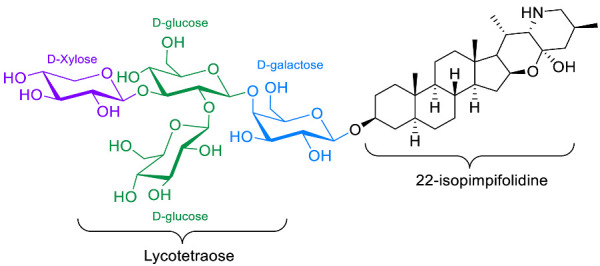	**Anti-inflammatory** [97]
**Solanidine**	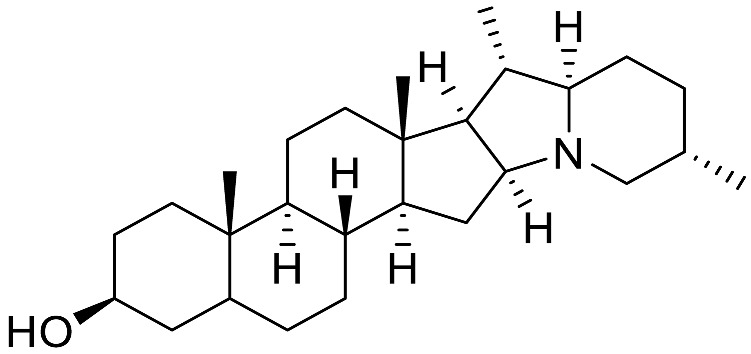	**Antiviral** (SARS-CoV-2 [70])

**Table 2 molecules-28-04957-t002:** SGAs and their respective association with cancer treatment in the last 10 years.

Compound	Cancer Type	Ref.
**α-Solanine**	Non small-cell lung cancer, hepatocellular carcinoma, choriocarcinoma, leukemia, esophageal, pancreatic, breast, adenocarcinoma, colorectal, endometrium, prostate, glioma, testicular, lung carcinoma, epithelial carcinoma	[98,99,100,101,102,103,104,105,106,107,108,109,110,111,112,113,114,115,116,117,118,119,120,121]
**Solamargine**	Nasopharyngeal carcinoma, prostate, multiple myeloma, lung cancer, leukemia, hepatoma, hepatocellular carcinoma, non-selective cytotoxicity, gastric	[122,123,124,125,126,127,128,129,130,131,132,133]
**Solasodine**	Ovarian	[134]
**α** **-Tomatine**	Ovarian	[135]
**Solasonine**	Gastric, pancreatic, bladder, leukemia, hepatocellular carcinoma	[123,132,136,137,138,139,140]
**Khasianine**	Leukemia	[123]
**α-Chaconine**	Endometrium, glioma	[100,104]
**Solanidine**	EAC solid tumor, CAM xenograft	[141]
**Solanindin**	Leukemia, lung, breast, colon	[91]
**Solalyraine A-E**	Lung carcinoma	[142]
**(25R)-22αN-4-nor-spirosol-5(6)-en-3β-ol-6-al glycosides**	Leukemia, histiocytic lymphoma, hepatocellular carcinoma	[92]
**16, 23-epoxy-22, 26-epimino-cholest22(N), 23, 25(26)-trien-3β-ol**	Gastric, hepatic	[143]
**Solanigroside P**	Gastric	[132]
**Lycoperoside H**	Colorectal	[144]

## Data Availability

Not applicable.

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
