# Peer review of "The Therapeutic Value of Solanum Steroidal (Glyco)Alkaloids: A 10-Year Comprehensive Review"

_molecules, 2023, doi:10.3390/molecules28134957_

Round 1

Reviewer 1 Report

Dear Authors,

The review is interesting, it has merit to be published. However its presentation needs to be improved. It is difficult to read. It is necessary to fully explain how the bibliographic review has been carried out. Scheme 1 is represented, out of context. In this sense, it is necessary to use tables in landscape format, and number the titles of the sections. There is a lot of room to improve the presentation. In some sections it uses bold font for certain compounds, in other sections it does not. I recommend following the style of Molecules, especially regarding how to cite and the references section.

Reviewer 2 Report

The paper tried to show the value of Solanum Steroidal (Glyco)Alkaloids in a 10-year comprehensive review.

The subject is worth to study, therefore the review is justified.

I found as very important the idea to show precise mechanisms of action of all SGAs in a single manuscript.

However, there are several issues that have to be clarified before paper publication.

Abstract – fonts – should be of the same kind.

Additionally, the abstract would benefit from more information about the potential practical implications of the research. The conclusion should underline the upcoming study direction, the knowledge gaps, etc.

They declared the review of the current literature (2012-2022). In the References the are also papers published in 1999.

Scheme 1 should be presented in Methodology section.

The text is very extensive. In many parts no literature is cited through ½ of the page. Therefore, it is not easy to read.

As most valuable in this paper I found the Table 1 information. Given in compact form.

For me, conclusions section it too long. It should be more compact with most important information.

Reviewer 3 Report

The bibliographical review is well structured. In my opinion, I think it gives a very partisan vision of the glycoalkaloids of the Solanaceae family. The toxic effects that some of these alkaloids have and that are present in vegetables and tubers so consumed cannot be omitted. by the world population such as potato, eggplant or tomato since they can mislead non-experts on the subject who can read this bibliographical review. If these vegetables are not cooked and handled properly, food poisoning can occur, which, although not very frequent, can have serious consequences for certain population groups.

For this reason, in my opinion, the authors should also include in this bibliographical review, not only the beneficial effects of some of these alkaloids of the nightshades. They should mention the toxic effects of other of these glycosidic alkaloids. In Spain there is a saying that says "All that it seems is not gold".

Round 2

Reviewer 1 Report

Dear Authors, the Ms was revised and improved. It´s ready to be accepted.

Author Response

We would like to thank reviewer 1 for its time and effort to make our manuscript better

Reviewer 2 Report

All my sugestions have been properly addressed.

Author Response

We would like to thank reviewer 2 for its insightful comments and help 

Reviewer 3 Report

In section 6 Toxicity & Therapeutic Challenges, I still miss the fact that the authors have not addressed the antinutritive effects derived from the glycosteroid alkaloids of Solanum.

Author Response

We agree and thank reviewer 3 for its comment pertraining to anti-nutrient properties. Anti-nutrient properties are known toxic effects of Solanum plant consumption. However, components other than SGAs are out of scope from our review article. Phytates, tannins, oxalates and saponins were not discussed in our main body of text and the addition of their toxic effects would seem out of place in a review focused on SGA bioactivities.

If however the editor judges that this inclusion would be valuable and pertinent to the review, we are ready to write a short paragraph on the subject.